# The linear ubiquitin chain assembly complex (LUBAC) generates heterotypic ubiquitin chains

Alan Rodriguez Carvajal[1], Irina Grishkovskaya[2], Carlos Gomez Diaz[1], Antonia Vogel[2], Adar Sonn-Segev[3], Manish S Kushwah[3], Katrin Schodl[1], Luiza Deszcz[1,2], Zsuzsanna Orban-Nemeth[2], Shinji Sakamoto[4], Karl Mechtler[2], Philipp Kukura[3], Tim Clausen[2], David Haselbach[2]*, Fumiyo Ikeda[1,5]*

[1]Institute of Molecular Biotechnology of the Austrian Academy of Sciences (IMBA), Vienna BioCenter (VBC), Vienna, Austria; [2]Research Institute of Molecular Pathology (IMP), Vienna BioCenter (VBC), Vienna, Austria; [3]Department of Chemistry, University of Oxford, Chemistry Research Laboratory, Oxford, United Kingdom; [4]Pharmaceutical Frontier Research Labs, JT Inc., Yokohama, Japan; [5]Medical Institute of Bioregulation (MIB), Kyushu University, Fukuoka, Japan

**Abstract** The linear ubiquitin chain assembly complex (LUBAC) is the only known ubiquitin ligase for linear/Met1-linked ubiquitin chain formation. One of the LUBAC components, heme-oxidized IRP2 ubiquitin ligase 1 (HOIL-1L), was recently shown to catalyse oxyester bond formation between ubiquitin and some substrates. However, oxyester bond formation in the context of LUBAC has not been directly observed. Here, we present the first 3D reconstruction of human LUBAC obtained by electron microscopy and report its generation of heterotypic ubiquitin chains containing linear linkages with oxyester-linked branches. We found that this event depends on HOIL-1L catalytic activity. By cross-linking mass spectrometry showing proximity between the catalytic RING-in-between-RING (RBR) domains, a coordinated ubiquitin relay mechanism between the HOIL-1-interacting protein (HOIP) and HOIL-1L ligases is suggested. In mouse embryonic fibroblasts, these heterotypic chains were induced by TNF, which is reduced in cells expressing an HOIL-1L catalytic inactive mutant. In conclusion, we demonstrate that LUBAC assembles heterotypic ubiquitin chains by the concerted action of HOIP and HOIL-1L.

*For correspondence:
david.haselbach@imp.ac.at (DH);
fumiyo.ikeda@bioreg.kyushu-u.ac.jp (FI)

## Introduction

Posttranslational modification of substrates with ubiquitin (ubiquitination) regulates a wide variety of biological functions. Ubiquitin forms chains via its seven internal Lys residues (Lys6, Lys11, Lys27, Lys29, Lys33, Lys48, and Lys63) through an isopeptide bond, or via Met1 through a peptide bond (*Komander and Rape, 2012*). The different ubiquitin chain types contribute to determine the fate of the substrate and biological outcomes regulated.

For substrate modification, ubiquitin can be conjugated through isopeptide bonds to Lys residues, thioester bonds formed with the side chain of Cys residues, or oxyester bonds formed with side chains of Ser and Thr residues (*Carvalho et al., 2007*; *McClellan et al., 2019*; *McDowell and Philpott, 2013*; *Vosper et al., 2009*; *Wang et al., 2007*; *Williams et al., 2007*). In addition, some bacteria have evolved a ubiquitination mechanism, carried out by proteins of the SidE effector family, that results in phosphoribosyl-linked ubiquitin conjugated to Ser residues of the protein substrate (*Shin et al., 2020*; *Bhogaraju et al., 2016*; *Qiu et al., 2016*). Ubiquitin also contains seven Thr (Thr7, Thr9, Thr12, Thr14, Thr22, Thr55, and Thr66) and three Ser (Ser20, Ser57, and Ser65) residues that could potentially act as sites for chain formation. Recently, ubiquitin polymerization through Ser or

Thr residues by a mammalian RING-in-between-RING (RBR)-type ubiquitin ligase, heme-oxidized IRP2 ubiquitin ligase 1 (HOIL-1L), has been described (*Kelsall et al., 2019*). The E3 ligase MYCBP2 was also shown to conjugate the ubiquitin to Thr residues through an ester bond (*Pao et al., 2018*). Besides these examples, all other known instances of ubiquitin-associated (UBA) oxyester bonds are found in the linkage between the ubiquitin and a non-ubiquitin substrate (*McClellan et al., 2019*; *McDowell and Philpott, 2013*).

HOIL-1L is a component of the linear ubiquitin chain assembly complex (LUBAC). LUBAC is thus far the only known E3 ubiquitin ligase complex that assembles linear/Met1-linked ubiquitin chains. LUBAC consists of two RBR-containing proteins: HOIL-1-interacting protein (HOIP) and HOIL-1L (*Kirisako et al., 2006*). LUBAC additionally contains the accessory protein Shank-associated RH domain-interacting protein (SHARPIN) (*Gerlach et al., 2011*; *Ikeda et al., 2011*; *Tokunaga et al., 2011*). HOIP has a catalytic centre in its RING2 domain responsible for assembly of linear ubiquitin chains, while HOIL-1L and SHARPIN have been recognized as accessory proteins for the process. It is more recent that HOIL-1L has been shown to catalyse ubiquitination (*Pao et al., 2018*; *Stieglitz et al., 2012*; *Smit et al., 2013*; *Tatematsu et al., 2008*). Linear ubiquitin chains and the three LUBAC components are essential components in biological functions including immune signalling (*Gerlach et al., 2011*; *Ikeda, 2015*; *Rahighi et al., 2009*; *Rittinger and Ikeda, 2017*; *Tokunaga et al., 2009*; *Iwai and Tokunaga, 2009*), development in mice (*Fujita et al., 2018*; *Peltzer et al., 2018*; *Peltzer et al., 2014*), protein quality control (*van Well et al., 2019*), Wnt signalling (*Rivkin et al., 2013*), and xenophagy (*Noad et al., 2017*; *van Wijk et al., 2017*). Therefore, it is important to understand how LUBAC assembles ubiquitin chains at the molecular level including how the catalytic activity of HOIL-1L contributes to the process.

In a recent study, Kelsall et al. demonstrated that recombinant HOIL-1L can polymerize ubiquitin via oxyester bonds on Ser and Thr. They also showed that a HOIL-1L C458S mutation in which a predicted ubiquitin-loading site is mutated results in reduction of oxyester-linked ubiquitination signals in cells suggesting their dependency on HOIL-1L (*Kelsall et al., 2019*). Moreover, Fuseya et al. recently demonstrated that HOIL-1L catalytic activity negatively regulates the TNF signalling cascade (*Fuseya et al., 2020*). However, understanding the precise mechanisms regulating this atypical form of ubiquitination and whether HOIL-1L as a part of LUBAC mediates this biochemical activity remains largely unresolved.

## Results

### Reconstitution and 3D reconstruction of LUBAC

We first set out to purify high-quality recombinant LUBAC for structural characterization and biochemical investigation. Purifications of the three LUBAC components expressed individually in *Escherichia coli* consistently gave low yields and isolated proteins were co-purified with several contaminants; this was particularly severe in purifications of full-length HOIP (*Figure 1A*). Given that HOIP is destabilized in cells lacking SHARPIN or HOIL-1L (*Gerlach et al., 2011*; *Ikeda et al., 2011*; *Tokunaga et al., 2011*; *Fujita et al., 2018*; *Peltzer et al., 2018*), we conjectured that HOIP could be unstable when recombinantly expressed in the absence of its interaction partners. To this end, we expressed HOIP (119.8 kDa), N-terminally His-tagged HOIL-1L (59.2 kDa), and N-terminally Strep (II)-tagged SHARPIN (43.0 kDa) in insect cells in order to co-purify the LUBAC holoenzyme by tandem affinity chromatography. Using this co-expression strategy, we were able to isolate three proteins of the expected molecular weights with no major contaminants as determined by SDS-PAGE followed by Coomassie staining (*Figure 1B*). Furthermore, we verified the identities of these proteins as the three LUBAC components by immunoblotting indicating the successful isolation of recombinant LUBAC (*Figure 1C*). Some truncation products of HOIP were detected by immunoblotting; however, these were not visible by Coomassie-based staining, indicating that they are not a prominent contaminant.

To examine the stability of the purified complexes, we performed gel filtration chromatography (*Figure 1—figure supplement 1A*). The elution profile of the complex contained two peaks eluting at 0.942 ml (peak I) and 0.972 ml (peak II) as well as one minor peak eluting at 1.158 ml (peak III). All of these peaks eluted earlier than the 158 kDa molecular weight standard, which eluted at 1.246 ml (*Figure 1—figure supplement 1C*). Given that the monomeric mass of purified LUBAC is expected

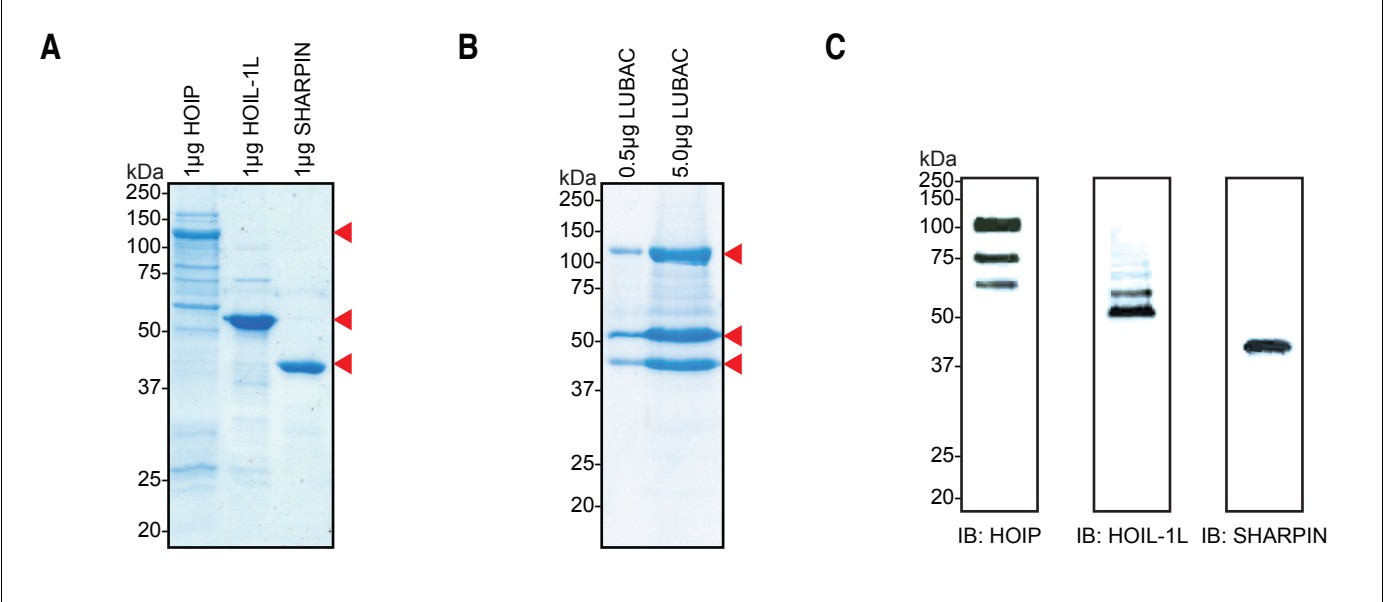

**Figure 1.** Co-expression and purification of linear ubiquitin chain assembly complex (LUBAC) yields high-quality protein. (**A**) SDS-PAGE analysis of individually purified LUBAC components. (**B**) SDS-PAGE analysis of co-expressed and purified LUBAC. (**C**) Immunoblot analysis of co-purified LUBAC. The online version of this article includes the following figure supplement(s) for figure 1:

**Figure supplement 1.** Gel filtration analysis of linear ubiquitin chain assembly complex (LUBAC) showing presence of multiple populations with different oligomeric states.

to be 222 kDa, the elution profile suggests that these peaks all correspond to assembled LUBAC in at least three populations of different oligomeric states. However, while peaks I and II contained all LUBAC components, peak III contained predominantly HOIL-1L and SHARPIN (*Figure 1—figure supplement 1A*, lower panel) indicating the presence of partially assembled complexes. To assess if this was a carryover from the purification or if the complex disassembles over time, we collected a fraction from peak II and reapplied it to the same column for a second isocratic elution (*Figure 1— figure supplement 1B*). The elution profile from this second tandem run contained almost exclusively peaks I and II, which correspond to assembled LUBAC (*Figure 1—figure supplement 1B*, lower panel). Conversely, peak III was almost entirely absent from the elution profile indicating that the complex is not prone to dissociation after purification.

To screen the homogeneity of the sample, we imaged fractions from peak II by negative staining electron microscopy. Micrographs show a monodisperse distribution of particles of similar size, which could be sorted into 2D class averages showing a distinct elongated dumbbell structure thus verifying the homogeneity of the sample (*Figure 2A* and *Figure 2—figure supplement 1A*). Furthermore, we were able to generate the first low-resolution 3D reconstruction of LUBAC from these particles (*Figure 2B* and *Figure 2—figure supplement 1B*). In accordance with the class averages, the model displays an elongated asymmetric crescent structure with the majority of the mass concentrated at one end. The class averages match calculated projections of the generated model very well showing that the model is self-consistent. (*Figure 2C* and *Figure 2—figure supplement 2*).

Collectively, we established a purification protocol to obtain high-purity and high-quality recombinant LUBAC that allowed us to generate the first low-resolution 3D reconstruction of the complex, from which we find the volume of structural envelope is approximately consistent with a 230 kDa particle.

## LUBAC exists as monomers and dimers with a 1:1:1 stoichiometry between HOIP, HOIL-1L, and SHARPIN

The precise stoichiometry and oligomerization state of LUBAC have not been established although recent structural work has suggested a 1:1:1 stoichiometry between the three core components (*Fujita et al., 2018*). To determine the stoichiometry and oligomerization state of LUBAC, we used

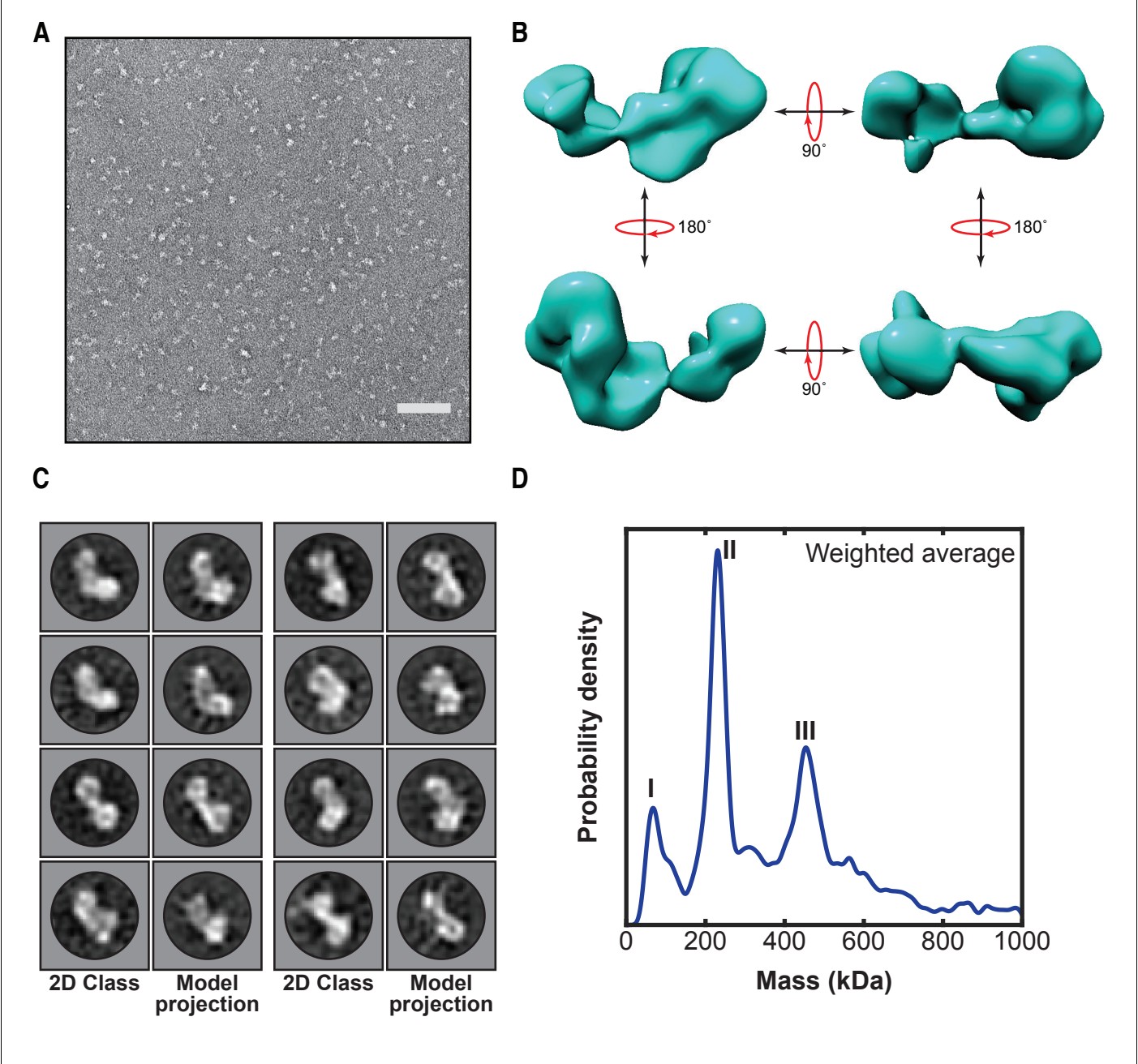

**Figure 2.** First low-resolution 3D map of linear ubiquitin chain assembly complex (LUBAC) obtained by negative staining electron microscopy of the recombinant complex. (**A**) Representative negative stain transmission electron micrograph of recombinant LUBAC. Scale bar: 100 nm. (**B**) 3D refined model of LUBAC obtained by single particle analysis of negative stained electron micrographs. (**C**) LUBAC 2D class averages matched to projections made from 3D refined map. (**D**) Mass photometry measurements of LUBAC indicate formation of a ternary complex with 1:1:1 stoichiometry that can form dimers.

The online version of this article includes the following figure supplement(s) for figure 2:

**Figure supplement 1.** Modelling of the linear ubiquitin chain assembly complex (LUBAC) by negative staining electron microscopy.

**Figure supplement 2.** Projections made from 3D refined model of linear ubiquitin chain assembly complex (LUBAC).

**Figure supplement 3.** Independent mass photometry measurements of linear ubiquitin chain assembly complex (LUBAC).

mass photometry (MP), a technique that enables accurate mass measurements based on the scattering of light by single macromolecules in solution (*Young et al., 2018*; *Sonn-Segev, 2019*; *Figure 2D* and *Figure 2—figure supplement 3*). MP measurements showed that the majority of species present in the samples had average masses of 231 kDa (peak II) and 454 kDa (peak III) (*Supplementary file 1*). An additional peak originating from a population with mass of less than 100 kDa was also measured (peak I), which could arise from isolated HOIL-1L or SHARPIN present in the measured sample.

With respect to the expected mass of the ternary LUBAC, the populations of peaks II and III nearly correspond with monomers (222 kDa) and dimers (444 kDa) of LUBAC with a 1:1:1 stoichiometry between HOIP, HOIL-1L, and SHARPIN.

## The RBR domains of HOIP and HOIL-1L are in close proximity

Understanding the precise mechanistic action of HOIL-1L and SHARPIN within LUBAC requires knowledge of structural and functional domain interactions between the three components. Current knowledge of the interaction domains of HOIP, HOIL-1L, and SHARPIN is shown in *Figure 3A*. Structural work of protein fragments has shown that HOIL-1L and SHARPIN interact with HOIP through their respective ubiquitin-like (UBL) domains, which bind cooperatively to the HOIP UBA domain (*Fujita et al., 2018*; *Liu et al., 2017*; *Yagi et al., 2012*; *Figure 3A*). By using truncation mutants, it has also been shown that the SHARPIN UBL domain interacts with the Npl zinc finger 2 (NZF2) domain of HOIP (*Ikeda et al., 2011*). More recently, structural work has revealed that SHARPIN and HOIL-1L interact with each other through their respective LUBAC-tethering motifs (LTMs) (*Fujita et al., 2018*). Otherwise, there is no further information available about the overall spatial arrangement of the domains of HOIP, HOIL-1L, and SHARPIN.

To obtain more detailed information about the spatial arrangement of LUBAC components, we performed cross-linking mass spectrometry (XL-MS) experiments. For this purpose, we used the amine-to-carboxyl-reactive cross-linker, 4-(4,6-dimethoxy-1,3,5-triazin-2-yl)-4-methylmorpholinium tetrafluoroborate (DMTMM), a zero-length cross-linker (*Leitner et al., 2014*) revealing Lys and Asp/Glu contacts that are adjacent to each other. As shown in *Figure 3B–E*, we observed an extensive network of intra-protein and inter-protein cross-links, providing a detailed picture of LUBAC assembly (*Figure 3B–E*, *Supplementary files 2* and *3*).

We detected some highly cross-linking residues that formed cross-links indiscriminately to all subunits of all proteins. These were HOIL-1L K174, SHARPIN K318, and HOIP K454/458. The high degree of cross-linking formed by these residues suggests that they are located on flexible regions. This notion is supported by the cleavage of HOIL-1L in the vicinity of K174 by MALT1, which suggests that the residue is located on a protease-accessible structure such as a flexible loop (*Elton et al., 2016*). The presence of flexible regions on the three LUBAC components may be related to the difficulties in determining the structures of full-length HOIP, HOIL-1L, and SHARPIN thus far.

We also observed DMTMM cross-links between LUBAC domains known to interact, such as between the HOIL-1L UBL and the SHARPIN UBL/LTM domains as well as between the HOIP UBA and HOIL-1L LTM domains, several cross-links pointed to new connections between the engaged proteins. Interestingly, we observed cross-links between the HOIL-1L RING1 and HOIP RING1/LDD domains, as well as between the HOIL-1L RING2 and the HOIP RING1/IBR/RING2/LDD domains, which suggest that the two enzymes have spatially connected catalytic activities. Additionally, HOIL-1L intra-protein cross-links were formed between its NZF domain and its RING1/IBR/RING2 domains, which could implicate the HOIL-1L NZF domain of unknown function in the catalytic action of HOIL-1L. In conclusion, the catalytic RBR domains of HOIP and HOIL1L seem to be close to each other, as well as the NZF and RBR regions of HOIL-1L. These data suggest that LUBAC may have a single catalytic centre containing the RBR domains of HOIP and HOIL-1L.

## Recombinant LUBAC assembles heterotypic ubiquitin chains containing linear and non-Lys linkages

To assess the ability of recombinant LUBAC to assemble linear ubiquitin chains, we performed in vitro ubiquitination assays. As expected, LUBAC generated linear ubiquitin chains in an ATP-dependent manner (*Figure 4A*). We also observed additional signals derived from co-purified LUBAC,

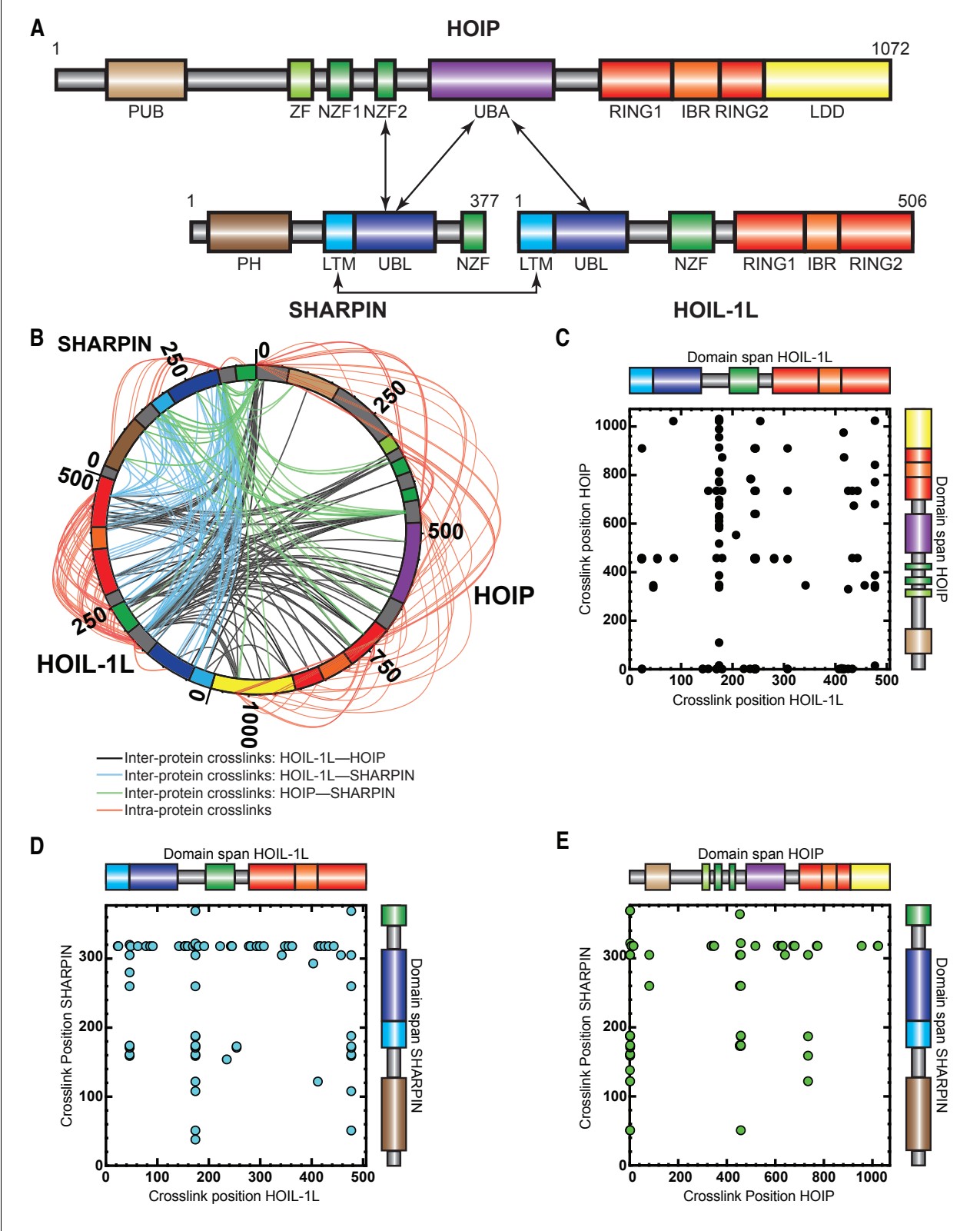

**Figure 3.** Cross-linking mass spectrometry (MS) analysis shows proximity between the catalytic domains of HOIL-1-interacting protein (HOIP) and heme-oxidized IRP2 ubiquitin ligase 1 (HOIL-1L). (A) Schematic representation of linear ubiquitin chain assembly complex (LUBAC) components with their domains and known interactions. (B) Circos plot of inter-protein cross-links formed between LUBAC components. (C) Detected inter-protein cross-links

*Figure 3 continued*
formed between HOIL-1L and HOIP. (**D**) Detected inter-protein cross-links formed between HOIL-1L and Shank-associated RH domain-interacting protein (SHARPIN). (**E**) Detected inter-protein cross-links formed between HOIP and SHARPIN.

which were absent from reactions containing the three individually purified and mixed LUBAC components (*Figure 4B*: red arrows).

The exclusive appearance of extra bands in the co-purified LUBAC sample indicates that the complex does not attain its full catalytic activity in vitro when the three LUBAC components are mixed. One possible explanation is that mixing of the three components is sufficient to activate HOIP but may not suffice for LUBAC to be adequately assembled. To determine if the mixed components could assemble the trimeric LUBAC, we carried out MP measurements of the three components mixed at an equimolar ratio (*Figure 4—figure supplement 2*). Our measurements showed a single prominent population of molecules with an average mass of 104 kDa. However, we were unable to detect any populations with the expected masses of 222 or 444 kDa corresponding to assembled and dimerized LUBAC. The absence of these populations indicates that mixing HOIP, HOIL-1L, and SHARPIN does not suffice to assemble LUBAC. We therefore conclude that the absence of bands in reactions carried out using mixed LUBAC components is a result of improper LUBAC assembly preventing the full catalytic activity of the complex.

Ubiquitin chains containing different linkages resolve at different apparent molecular masses by SDS-PAGE separation (*Emmerich and Cohen, 2015*). Therefore, we hypothesized that the two bands in each pair contained different ubiquitin linkages. To examine the presence of different Lys-linked bonds in the heterotypic chains, we carried out in vitro ubiquitination assays using different ubiquitin Lys to Arg (KR) mutants (*Figure 4C*). Heterotypic ubiquitin chain assembly was observed with all the tested mutants, K6R, K11R, K27R, K29R, K33R, K48R, and K63R. To rule out the possibility that mutation of a single Lys could be compensated by ubiquitination of a separate Lys residue, we also performed in vitro ubiquitination assay using ubiquitin lacking all Lys residues ($K_0$) (*Figure 4D*). Despite reduced reaction efficiency, the $K_0$ mutant could also be used by LUBAC to generate the heterotypic ubiquitin chains. At present we cannot account for the apparent difference in the ratio between branched tri- and tetra-ubiquitin chains observed across different experiments, and which is particularly pronounced with the $K_0$ substrate. However, we speculate that these differences arise from varied reaction efficiencies. We propose this is the case given that branched tri-ubiquitin is the more prominent species when $K_0$ ubiquitin is used. With this mutant we also observe a drastic loss of overall reaction efficiency with respect to reactions containing wild-type (WT) ubiquitin. Nevertheless, both branched chains could be detected to some extent across all replicates of all experiments.

To further analyse the linkage types of ubiquitin chains, we performed ubiquitin chain restriction (UbiCRest) (*Hospenthal et al., 2015*) on the ubiquitin chains generated by LUBAC (*Figure 4E*). UbiCRest using linkage-specific deubiquitinases (DUBs) allows detection of specific ubiquitin chains by loss of signals when the linkage is targeted. Interestingly, the additional signals from ubiquitin chains assembled by LUBAC disappeared by treatment with Cezanne, a DUB specific for the Lys11 linkage (*Bremm et al., 2010*); or vOTU, a DUB targeting Lys linkages (*Akutsu et al., 2011*; *Figure 4E*; upper red arrow). The linear linkage-specific DUB OTULIN (*Fiil et al., 2013*) hydrolysed all linear ubiquitin bonds (*Figure 4E*: upper panel), corresponding to most of the ubiquitin signal, but left non-linear di- and tri-ubiquitin residues (*Figure 4E*: lower panel).

These data collectively indicate that recombinant LUBAC assembles heterotypic ubiquitin chains containing predominantly linear linkage with non-Lys-linked branches.

## LUBAC assembles heterotypic poly-ubiquitin chains containing linear and ester-linked bonds in vitro and in cells

A recent study showed that recombinant HOIL-1L can generate di-ubiquitin linked via an oxyester bond in vitro and can also ubiquitinate substrates through oxyester bonds in cells (*Kelsall et al., 2019*). Therefore, we tested for the presence of oxyester bonds in the ubiquitin chains by checking their sensitivity to the nucleophile hydroxylamine (*Figure 5A*). Treatment with hydroxylamine resulted in the disappearance of the upper band of the linear tetra-ubiquitin chain (*Figure 5A*: red

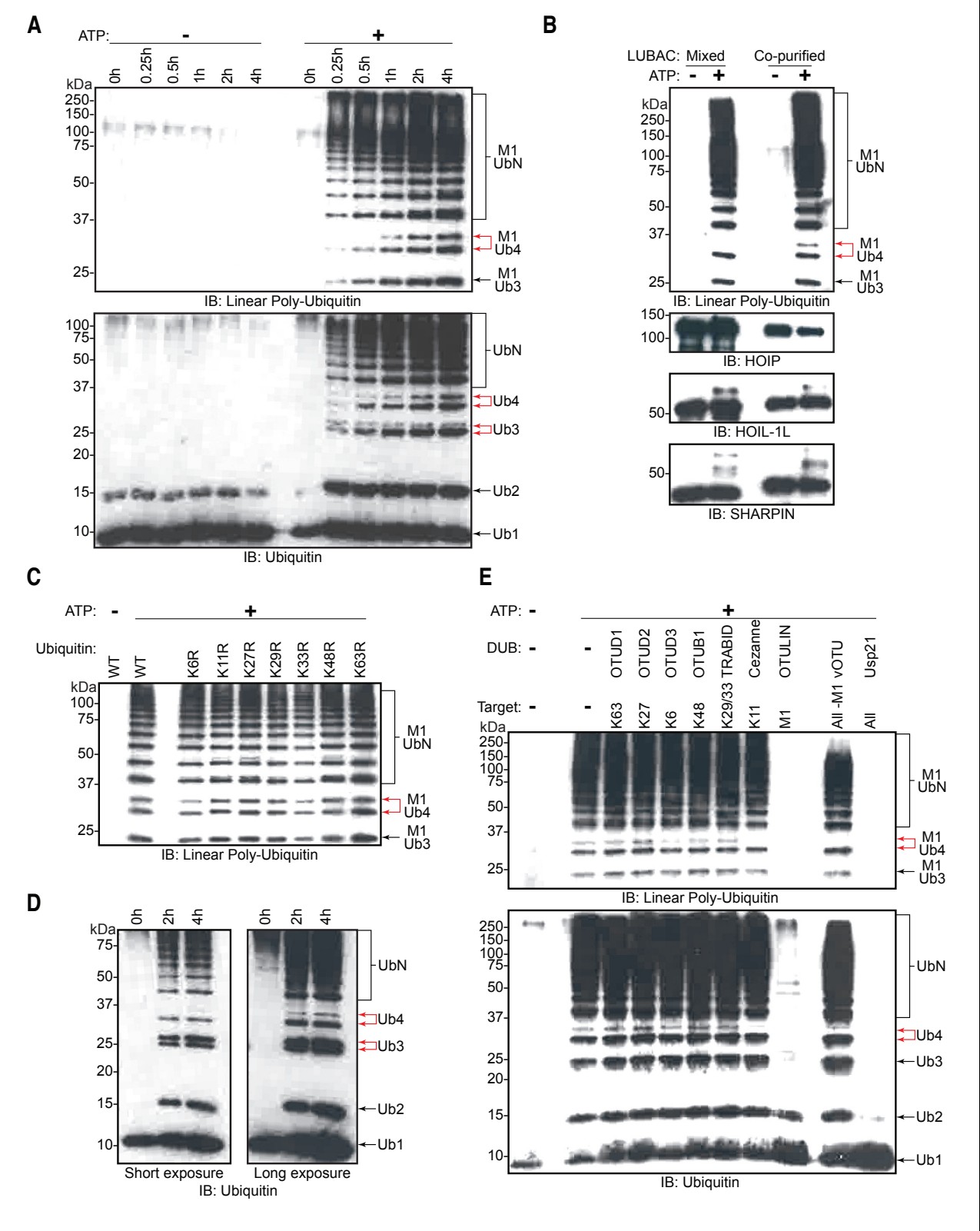

**Figure 4.** Linear ubiquitin chain assembly complex (LUBAC) assembles heterotypic poly-ubiquitin chains containing M1 and non-Lys linkages in vitro. (**A**) Time course of co-purified LUBAC in vitro ubiquitin chain assembly reaction. (**B**) Comparison of in vitro chain assembly between HOIL-1-interacting protein (HOIP), heme-oxidized IRP2 ubiquitin ligase 1 (HOIL-1L), and Shank-associated RH domain-interacting protein (SHARPIN) mixed at 1:1:1 molar ratio versus co-purified LUBAC. (**C**) LUBAC in vitro chain assembly using different ubiquitin K to R mutants. (**D**) LUBAC in vitro chain assembly using K0

*Figure 4 continued on next page*

*Figure 4 continued*

ubiquitin. (E) Ubiquitin chain restriction (UbiCRest) analysis of poly-ubiquitin chains assembled by LUBAC in vitro. All experiments were performed in triplicate representative results are shown.

The online version of this article includes the following figure supplement(s) for figure 4:

**Figure supplement 1.** Anti-linear ubiquitin antibody validation.

**Figure supplement 2.** Independently purified HOIL-1-interacting protein (HOIP), heme-oxidized IRP2 ubiquitin ligase 1 (HOIL-1L), and Shank-associated RH domain-interacting protein (SHARPIN) mixed at an equimolar ratio cannot reconstitute the trimeric linear ubiquitin chain assembly complex (LUBAC).

arrow). This indicates that the chain branching is achieved by formation of an oxyester bond between ubiquitin moieties similar to the observation in the previous study (*Kelsall et al., 2019*).

To further ascertain whether LUBAC assembles ester-linked ubiquitin polymers, we carried out in vitro ubiquitination assays using N-terminally His$_6$-tagged ubiquitin, which cannot be used to assemble linear ubiquitin chains (*Kirisako et al., 2006*). LUBAC inefficiently assembled di- and tri-ubiquitin chains with His-ubiquitin, which are sensitive to hydroxylamine treatment (*Figure 5B*). These results demonstrate that when the N-terminus of ubiquitin is not available, LUBAC can still assemble oxyester-linked poly-ubiquitin even in the absence of linear ubiquitin chains.

To determine if LUBAC was generating any other kind of ubiquitin linkage other than linear and oxyester bonds, we treated LUBAC-assembled ubiquitin chains with OTULIN and hydroxylamine (*Figure 5—figure supplement 1A*). Restriction with OTULIN degraded the majority of the signal from ubiquitin chains leaving short di- and tri-ubiquitin polymers. Subsequent treatment with hydroxylamine resulted in complete degradation of the poly-ubiquitin signals. These results indicate that LUBAC assembles exclusively OTULIN-sensitive linear and hydroxylamine-sensitive oxyester linkages between ubiquitin moieties.

In our UbiCRest analysis we observed cleavage of the oxyester-linked branches from the LUBAC-assembled poly-ubiquitin chains by Cezanne and vOTU (*Figure 4E*). Therefore, we also tested the sensitivity of the exclusively oxyester-linked ubiquitin chains assembled by LUBAC with His$_6$-ubiquitin to the two DUBs (*Figure 5—figure supplement 1B*). Both Cezanne and vOTU could fully degrade these short oxy-ester ubiquitin polymers, which is in agreement with a recent study reporting the ester-directed activity of the two DUBs (*De Cesare et al., 2021*).

Ubiquitin contains seven Thr residues, three Ser residues, and one Tyr residue, which could theoretically act as sites for ester bond formation. Therefore, we aimed to identify the positions of the ester-linked branches using mass spectrometry by searching for GG dipeptides covalently attached through oxyester bonds to Ser, Thr, or Tyr, as was previously done by *Kelsall et al., 2019*. To this end, we analysed ubiquitin chains formed by LUBAC by LC-MS. MS/MS spectra of GG-conjugated dipeptides at residues Thr12 and Thr55 of ubiquitin were detected from these samples, in which there was complete coverage of the ubiquitin amino acid sequence 1–73 (*Figure 5C*). Structural analysis shows that these two residues are positioned at opposite sides of ubiquitin and neither of them is located in proximity to Met1 or the C-terminal Gly76 (*Figure 5D*; PDB:1UBI). This suggests that branches could potentially exist on both sites of a single ubiquitin molecule located at any position of a linear ubiquitin chain without creating any steric hindrances.

To further investigate if Thr12 and Thr55 are the sites of oxyester bond formation, we performed in vitro ubiquitination assays using ubiquitin mutated at Thr12 and/or Thr55 to Val (*Figure 5E*). Individual or concomitant mutation of Thr12 and Thr55 (Ub T$_{12}$V, Ub T$_{55}$V, or Ub T$_{12/55}$V, respectively) did not result in loss of chain branching. These results suggest that linear chain branching is not a strictly site-specific event. We next probed the existence of the hybrid chains in cells. It has recently been reported that these chains are formed on IRAK1, IRAK2, and Myd88 in response to activation of TLR signalling (*Kelsall et al., 2019*). Given that LUBAC is involved in the TNF signalling cascade, we examined the induction of these heterotypic ubiquitin chains in TNF-treated cells. Linear ubiquitin chains were enriched by GST pull-down using the NEMO-UBAN and zinc finger domains (NEMO$_{250-412}$) from total cell extracts of TNF stimulated mouse embryonic fibroblasts (MEFs). In this set-up, the GST-NEMO-UBAN was employed as a ubiquitin-binding matrix to enrich linear-linked chains and not as a substrate for LUBAC. The enriched poly-ubiquitin chains were then tested for hydroxylamine sensitivity (*Figure 6A*). TNF stimulation increased formation of linear ubiquitin chains detected by immunoblotting using an antibody specific for linear ubiquitin chains (*Figure 6A*: PD

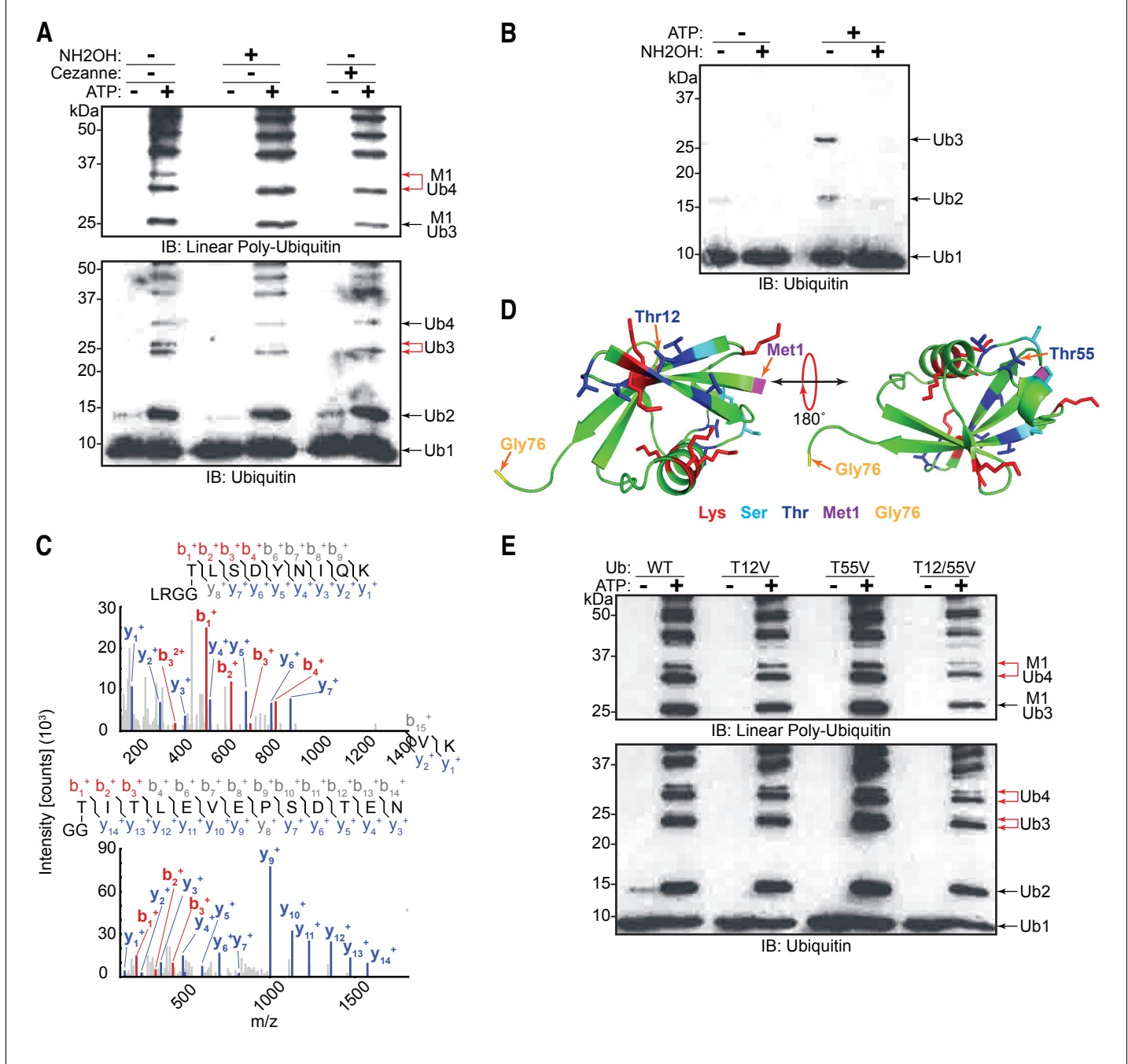

**Figure 5.** Linear ubiquitin chain assembly complex (LUBAC) assembles heterotypic poly-ubiquitin chains containing M1 and ester bond linkages at T12 and T55. (**A**) Treatment of LUBAC-assembled heterotypic poly-ubiquitin chains with hydroxylamine. (**B**) Hydroxylamine treatment of ubiquitin polymers assembled by LUBAC using N-terminally blocked ubiquitin. (**C**) MS/MS spectra of ubiquitin polymerized at T55 (top) and T12 (bottom). Poly-ubiquitin chains assembled by LUBAC were separated by SDS-PAGE, bands were cut from the gel and subjected to mass spectrometry analysis. (**D**) Positions of Thr12 and Thr55 on structure of ubiquitin (PDB:1UBI). (**E**) Assembly of ubiquitin chains by LUBAC using different ubiquitin Thr to Val point mutants as substrates. All experiments were performed in triplicate representative results are shown.

The online version of this article includes the following figure supplement(s) for figure 5:

**Figure supplement 1.** Cezanne, vOTU, and hydroxylamine can cleave oxyester bonds in linear ubiquitin chain assembly complex (LUBAC)-assembled heterotypic ubiquitin chains.

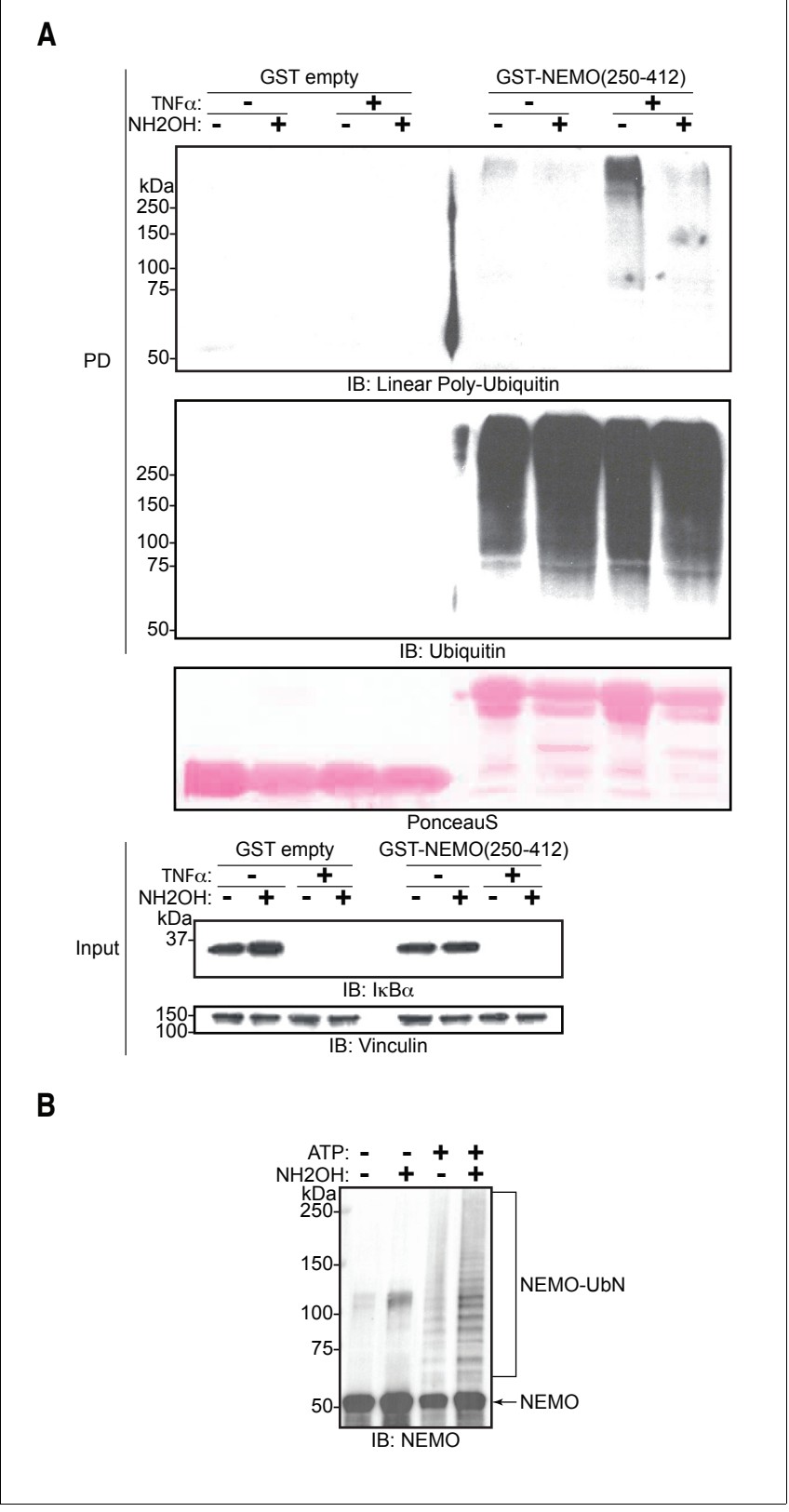

**Figure 6.** M1-linked/linear ubiquitin chains generated in cells and conjugated to NEMO in vitro are sensitive to hydroxylamine treatment. (A) Hydroxylamine treatment of M1 linkage-containing poly-ubiquitin chains assembled in response to TNF in wild-type (WT) mouse embryonic fibroblasts (MEFs). MEFs were treated with TNF for 15 min

*Figure 6 continued on next page*

*Figure 6 continued*
and lysed, lysates were subjected to GST PD using GST or GST-NEMO(250-412), beads were treated with buffer or hydroxylamine for 30 min, bound ubiquitin species were then analysed by immunoblotting. (B) Treatment of linear ubiquitin chain assembly complex (LUBAC)-dependent NEMO in vitro ubiquitination reactions with hydroxylamine. All experiments were performed in triplicate representative results are shown.

top panel lane 7). These chains proved sensitive to hydroxylamine treatment (*Figure 6A*: PD top panel lane 8). In contrast, the signals from the overall population of ubiquitin chains are unchanged by either TNF stimulation or hydroxylamine treatment (*Figure 6A*: PD bottom panel lanes 5–8). Together these results suggest that ubiquitin chains containing both linear linkages and hydroxylamine-sensitive ester bonds are produced in cells in response to TNF stimulation.

To determine if substrate-conjugated linear-linked ubiquitin chains are also branched, we carried out an in vitro ubiquitination assay using NEMO as the substrate then subjected the products to hydroxylamine treatment (*Figure 6B*). Treatment of ubiquitinated NEMO with the compound resulted in enrichment of some lower molecular weight bands corresponding to ubiquitinated NEMO when blotting with a NEMO-specific antibody. However, the majority of the signal from the ubiquitinated NEMO was unperturbed. These results suggest that the conjugation of ubiquitin chains to the substrate is principally through hydroxylamine-resistant isopeptide bonds with some hydroxylamine-sensitive chain branching or substrate ubiquitination also taking place on longer linear chains.

## HOIP assembles linear ubiquitin chains that are subsequently branched with ester bonds by HOIL-1L

We next proceeded to probe how HOIP and HOIL-1L generate these heterotypic ubiquitin chains. HOIP specifically assembles linear ubiquitin chains through the action of its RBR domains wherein Cys885 is the catalytic residue (*Smit et al., 2012*; *Stieglitz et al., 2013*); similarly HOIL-1L catalyses ester bond-directed ubiquitination via its RBR domain where Cys460 is the catalytic site (*Kelsall et al., 2019*; *Smit et al., 2013*). Therefore, we examined the possibility that HOIP assembles a linear ubiquitin chain, which is subsequently branched with ester bonds by HOIL-1L. To this end, we purified different LUBAC containing catalytically inactive HOIP (HOIP C885A), HOIL-1L (HOIL-1L C460A), or both HOIP C885A and HOIL-1L C460A. Mutation of these residues did not impair the purification of LUBAC, and all variants of the complex could be isolated at similar yields and to similar degrees of purity (*Figure 7—figure supplement 1*). In line with our hypothesis, LUBAC containing HOIL-1L C460A generated chains with linear linkages, yet the double band indicative of heterotypic chain assembly was absent (*Figure 7A*; lane 6 upper red arrow). These observations indicate that HOIL-1L is responsible for the formation of the ester-linked branches. Consistent with previous reports (*Smit et al., 2013*; *Fuseya et al., 2020*), HOIP assembled longer linear ubiquitin chains in the absence of HOIL-1L catalytic activity and did so more rapidly (*Figure 7A*: lanes 2 and 6). Conversely, LUBAC containing HOIP C885A was incapable of polymerizing ubiquitin altogether (*Figure 7A*; lanes 4 and 8) as expected. These results suggest that HOIL-1L catalytic activity disturbs linear ubiquitin chain formation but requires HOIP catalytic activity.

The gel-migration pattern differed between ubiquitin chains assembled by LUBAC with HOIL-1L C460A and the WT complex (*Figure 7A*: lanes 2 and 6). Therefore, we compared the presence of non-linear bonds in the chains assembled by the different complexes by OTULIN. The restriction analyses showed that the ubiquitin chains assembled by LUBAC contained non-linear di- and tri-ubiquitin chains, whereas chains assembled by HOIL-1L C460A-containing LUBAC generated exclusively OTULIN-sensitive linear ubiquitin chains (*Figure 7B*). These results further support the claim that chain branching is dependent on the catalytic action of HOIL-1L while HOIP exclusively assembles linear ubiquitin chains.

## HOIL-1L catalytic activity is required for branched ubiquitin chain formation in cells

To further study the catalytic function of HOIL-1L in cells, we used cells derived from a HOIL-1L C458A knock-in mouse (*Rbck1^(C458A/C458A)*) generated by CRISPR/Cas9 gene-editing technology (*Figure 7—figure supplement 2A–C*). The expression levels of the three LUBAC components are similar

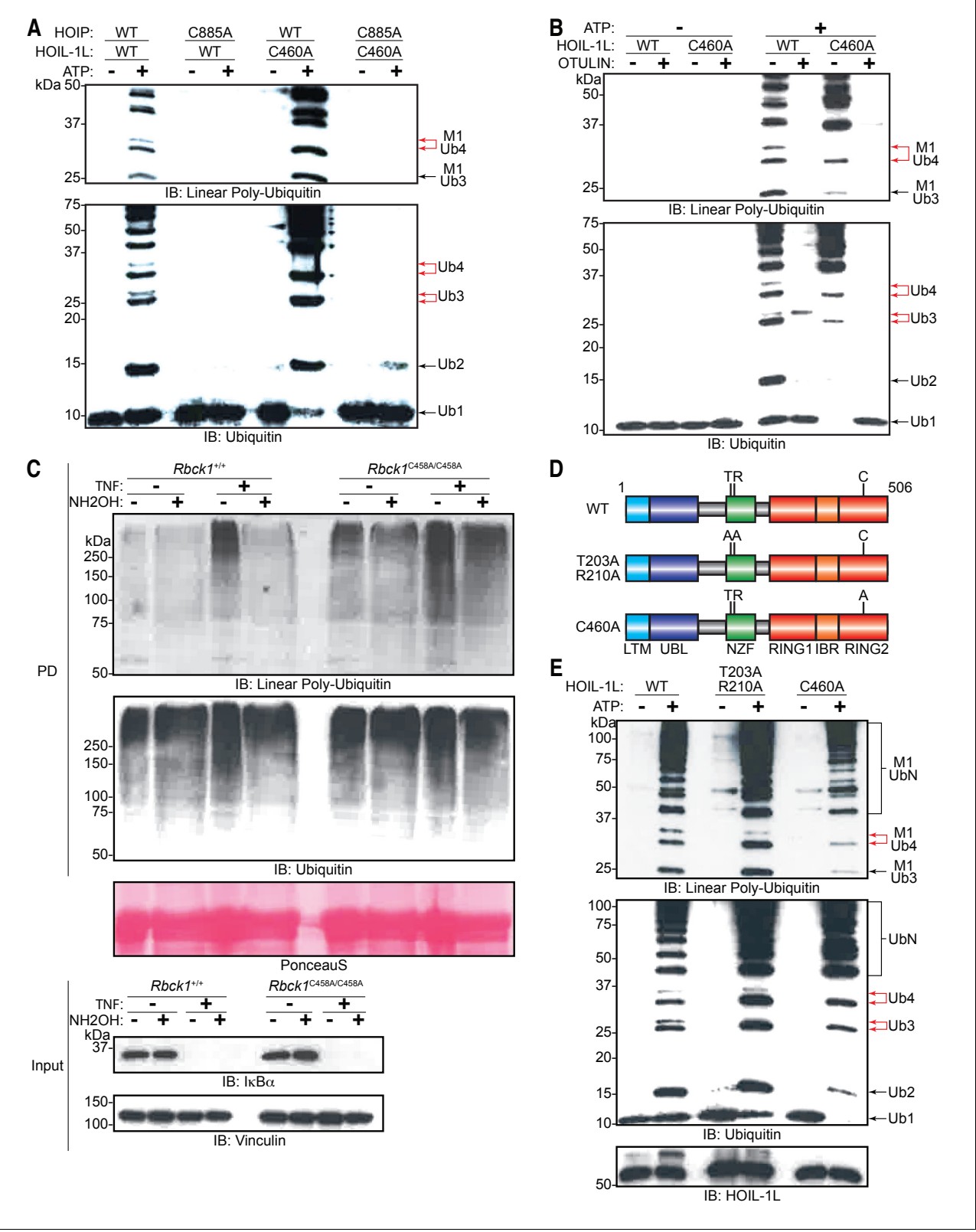

**Figure 7.** Heme-oxidized IRP2 ubiquitin ligase 1 (HOIL-1L) generates ester linkages on heterotypic chains but requires HOIL-1-interacting protein (HOIP) catalytic activity to polymerize ubiquitin. (**A**) Comparison of linear ubiquitin chain assembly complex (LUBAC) in vitro chain assembly by complexes containing different catalytically inert mutants of HOIP and HOIL-1L. (**B**) OTULIN restriction of poly-ubiquitin chains assembled by LUBAC containing wild-type (WT) or catalytically inert HOIL-1L. (**C**) Hydroxylamine treatment of M1 linkage-containing poly-ubiquitin chains assembled in

*Figure 7 continued on next page*

*Figure 7 continued*

response to TNF in WT and *Rbck1*$^{C458A/C458A}$ mouse embryonic fibroblast (MEF) cells. Cells were treated with TNF (50 ng/ml) for 15 min and lysed, lysates were subjected to GST PD using GST-NEMO (250-412), beads were treated with buffer or hydroxylamine for 30 min, bound ubiquitin species were then analysed by immunoblotting. (D) Schematic representations of different HOIL-1L mutants. (E) Comparison of LUBAC in vitro chain assembly by complexes containing different HOIL-1L mutants. All experiments were performed in triplicate representative results are shown.

The online version of this article includes the following figure supplement(s) for figure 7:

**Figure supplement 1.** Purification of linear ubiquitin chain assembly complex (LUBAC) containing catalytically inert HOIL-1-interacting protein (HOIP) and heme-oxidized IRP2 ubiquitin ligase 1 (HOIL-1L) proteins.
**Figure supplement 2.** Generation of *Hoil-1l*$^{C458A/C458A}$ mice.

in MEFs in *Rbck1*$^{+/+}$ and *Rbck1*$^{C458A/C458A}$ MEFs (*Figure 7—figure supplement 2D*). In agreement with numerous studies from the past, two species of HOIL-1L were detected in *Rbck1*$^{+/+}$ MEFs by immunoblotting; the slower migrating species originating from auto-ubiquitinated HOIL-1L was absent in *Rbck1*$^{C458A/C458A}$ MEFs.

To determine if HOIL-1L is the responsible ligase for linear ubiquitin chain branching during TNF signalling, TNF-treated *Rbck1*$^{C458A/C458A}$ MEF lysates were subjected to GST pull-down with GST-NEMO$_{250-412}$ followed by hydroxylamine treatment (*Figure 7C*). Similarly, to WT MEFs, TNF stimulation led to an increase in the levels of linear ubiquitin chains in *Rbck1*$^{C458A/C458A}$ MEFs (*Figure 7C*: PD top panel lanes 3 and 7). However, the enriched ubiquitin signals detected by immunoblotting revealed a higher basal level of linear ubiquitin chains in *Rbck1*$^{C458A/C458A}$ MEFs compared to those in *Rbck1*$^{+/+}$ MEFs (*Figure 7C*: PD top panel lanes 1 and 5). Moreover, unlike in *Rbck1*$^{+/+}$ MEFs, linear ubiquitin chains in *Rbck1*$^{C458A/C458A}$ MEFs were not sensitive to degradation by hydroxylamine treatment (*Figure 7C*: PD bottom panel). However, there was no obvious difference when blotting for general ubiquitin between *Rbck1*$^{+/+}$ and *Rbck1*$^{C458A/C458A}$ MEFs regardless of TNF stimulation or chain hydroxylamine treatment (*Figure 7C*: PD bottom panel). These results show that ester-linked branching of linear ubiquitin chains formed during TNF signalling in MEFs is dependent on the catalytic activity of HOIL-1L.

## HOIL-1L Npl4 zinc finger is involved in the formation of branching ubiquitin chains in vitro

Since the catalytic action of HOIP precedes that of HOIL-1L and the assembly of linear ubiquitin chains precedes the appearance of the branches, we hypothesized that HOIL-1L interacts with a linear ubiquitin chain as a substrate for branching via its linear ubiquitin chain-specific binding domain NZF (*Sato et al., 2011*). To test this possibility, we purified LUBAC containing a mutant in which critical residues for linear ubiquitin chain recognition are mutated (HOIL-1L T203A,R210A) (*Figure 7D*). In agreement with our hypothesis, chain branching activity by LUBAC was partially impaired by the HOIL-1L T203A,R210A mutant when compared to the WT or HOIL-1L C460A (*Figure 7E*). Interestingly, LUBAC with HOIL-1L T203A,R210A assembled ubiquitin chains more efficiently than WT LUBAC but less efficiently than HOIL-1L C460 (*Figure 7E*).

These data collectively show that HOIP assembles linear ubiquitin chains, which are subsequently branched by HOIL-1L in a process involving its NZF domain and which requires the catalytic activity of HOIP.

In summary, we identified that LUBAC assembles heterotypic linear/ester-linked poly-ubiquitin chains in vitro and in cells in response to TNF stimulation. We also show that these chains are synthesized through the concerted action of HOIP and HOIL-1L (*Figure 8*). These chains may contribute in modulating the speed and/or efficiency of linear ubiquitin chain synthesis by LUBAC.

## Discussion

We present here the first 3D reconstruction of the ternary LUBAC. We cannot determine the exact positions of HOIP, HOIL-1L, and SHARPIN in the map at the current resolution; however, our map is the first structure encompassing the LUBAC holoenzyme. We also made other novel structural observations that LUBAC exists as monomers and dimers of a ternary complex with 1:1:1 stoichiometry between HOIP, HOIL-1L, and SHARPIN. This is in agreement with observations made in recent structural work (*Fujita et al., 2018*). We also present new data addressing the question of LUBAC

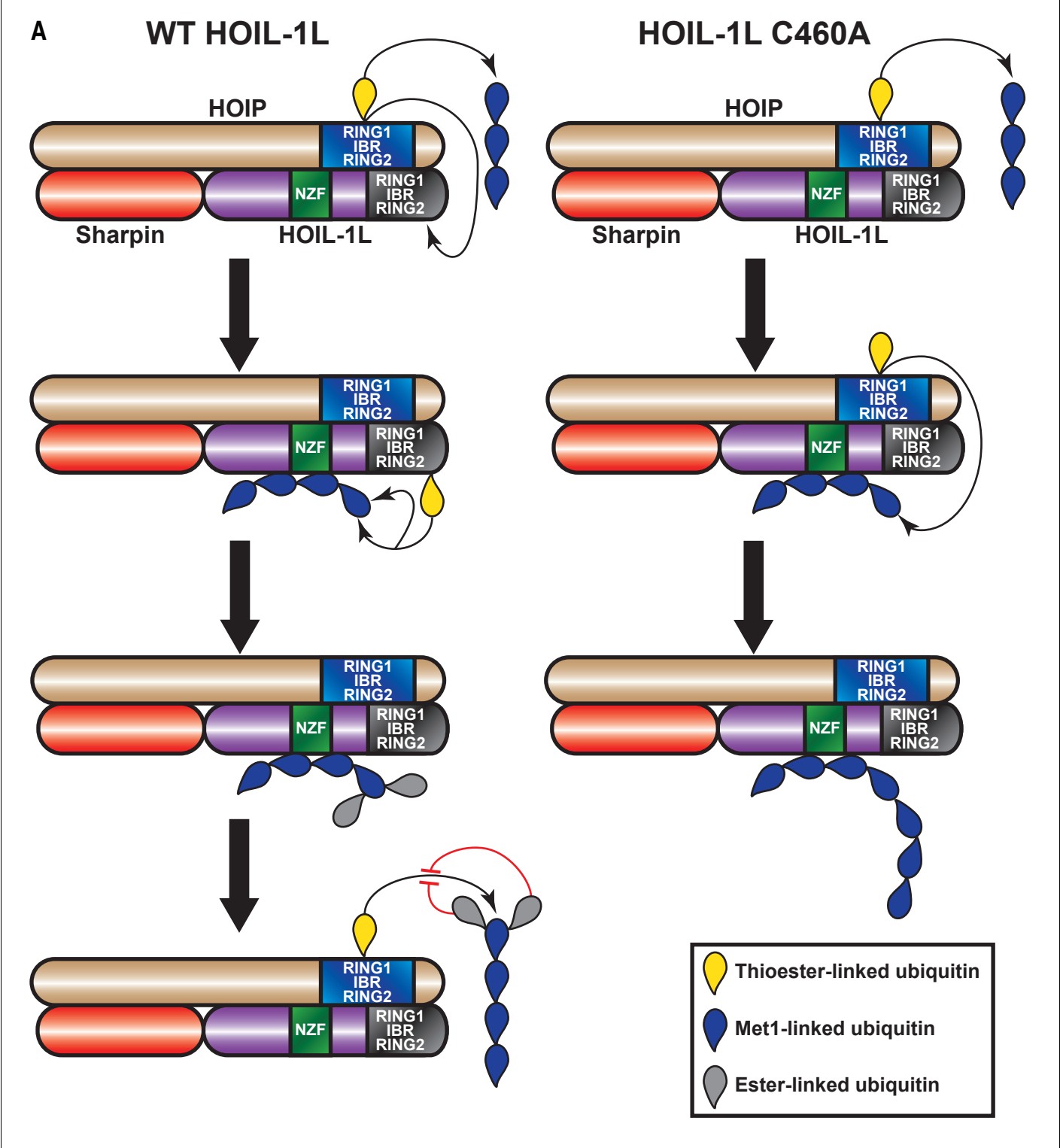

**Figure 8.** Proposed model for concerted action between HOIL-1-interacting protein (HOIP) and heme-oxidized IRP2 ubiquitin ligase 1 (HOIL-1L) in heterotypic chain assembly. HOIP and HOIL-1L assemble heterotypic chains through a Cys relay mechanism. HOIP forms a thioester bond to ubiquitin, which can be either transferred to a thioester bond on HOIL-1L or added to a nascent linear ubiquitin chain. HOIL-1L subsequently binds the linear ubiquitin chain through its Npl zinc finger (NZF) domain and branches it with ester linkages. The resulting heterotypic poly-ubiquitin chains contain predominantly linear linkages with ester-linked branches.

oligomerization. Early gel filtration analysis of cell-derived LUBAC suggested formation of a large or oligomeric complex with a molecular mass of over 600 kDa (*Kirisako et al., 2006*). However, our MP measurements using recombinant LUBAC show that this is not the case in vitro. This could be due to LUBAC in cells interacting with cellular components such as SPATA2 and CYLD (*Wagner et al., 2016*; *Schlicher et al., 2016*; *Kupka et al., 2016*; *Elliott et al., 2016*). This discrepancy would also be accounted for by a particle of non-globular structure, which would elute earlier than expected from a gel filtration column (*Siegel and Monty, 1966*; *Erickson, 2009*). Accordingly, the 3D map we obtained from negative stain electron micrographs shows a particle of elongated structure.

We propose a catalytic Cys relay mechanism in LUBAC: HOIP receives ubiquitin from the E2 and uses this ubiquitin to assemble linear chains or transfers it to HOIL-1L, which then branches the chains assisted by its NZF domain (*Figure 8*). One recent example of such a mechanism is the E3 ligase MYCBP2, which also conjugates the ubiquitin to Thr residues through an ester bond (*Pao et al., 2018*). A relay mechanism would require spatial proximity between the catalytic sites of the two ligases. Our XL-MS data show that this is indeed the case based on cross-linking between the RBR domains of HOIP and HOIL-1L. We also detected proximity between the HOIL-1L NZF and the RBR domains of both HOIP and HOIL-1L, which could contribute to the branching of the linearly linked ubiquitin dimer. Collectively, these observations suggest that LUBAC has a single catalytic centre containing the RBR domains of HOIP and HOIL-1L.

HOIP and HOIL-1L have distinct catalytic behaviour. HOIP exclusively targets Met1 of a ubiquitin moiety for polymerization (*Swatek et al., 2019*); in contrast, HOIL-1L seems to have more flexibility with target sites in forming ester bonds. We identified Thr12 and Thr55 in ubiquitin as target sites by mass spectrometry; yet individual and concomitant mutation of these residues did not abolish heterotypic chain assembly suggesting existence of secondary sites. Indeed, in ubiquitin, Thr12 is structurally located in the vicinity of Thr7, Thr9, and Thr14, whereas Thr55 is located near Thr22, Ser20, and Ser57. A recent study identified Thr12, Ser20, and Thr22, but not Thr55, as target sites by isolated HOIL-1L (*Kelsall et al., 2019*). These differences may depend on HOIL-1L as an isolated protein, or HOIL-1L as a part of LUBAC where HOIP and SHARPIN contribute to site selection.

We speculate that HOIP and SHARPIN contribute to the catalytic action of HOIL-1L based on the observed loss of the HOIL-1L auto-ubiquitination signal in cells derived from SHARPIN-deficient mice (*Gerlach et al., 2011*; *Ikeda et al., 2011*; *Tokunaga et al., 2011*) and HOIP knockout mice (*Peltzer et al., 2014*). This effect is identical in cells derived from HOIL-1L C458S mice (*Kelsall et al., 2019*), as well as HOIL-1L C458A mice from this study. Investigating the precise mechanism how the non-catalytic LUBAC component SHARPIN contributes to the HOIL-1L catalytic activity may shed light on its role in the complex.

Our analysis of chain branching on substrate-conjugated ubiquitin chains suggests that the oxyester branch points of the chain are located distally from the substrate. Treatment of NEMO in vitro ubiquitination reactions resulted in some change of signal, but the overall levels of ubiquitinated substrate were unperturbed, indicating that linear ubiquitin chains are not conjugated onto NEMO via oxyester bonds. This is consistent with a previous report showing that loss of HOIL-1L catalytic activity impairs but does not prevent conjugation of linear ubiquitin chains to the LUBAC substrates IRAK1, IRAK2, and MyD88 (*Kelsall et al., 2019*). Our results would also suggest that HOIL-1L is not responsible for priming substrates with a mono-ubiquitin that is then extended into an linear ubiquitin chain by HOIP. However, based on our data, we cannot exclude the possibility that HOIL-1L also has Lys-directed ubiquitin conjugation capabilities as has been suggested by prior studies (*Smit et al., 2013*; *Fuseya et al., 2020*).

Our data suggests that linear ubiquitin chains are heavily branched in cells. Hydroxylamine treatment results in a near complete loss of signal from linear ubiquitin chains in cells, which suggests that linear chains are heavily branched. In contrast, our in vitro data suggests that only small populations of linear chains are successfully branched. One explanation could be that there are other factors present in cells that assist HOIL-1L in the process of chain branching and which remain hitherto unidentified.

DUBs targeting ester linkages remain to be identified. Our data show that a fragment of Cezanne, a DUB specific for Lys11 linkages (*Bremm et al., 2010*; *Mevissen et al., 2016*), can cleave the ester-linked branches in vitro. Cezanne is a known negative regulator of NF-κB (*Enesa et al., 2008*; *Evans et al., 2003*; *Evans et al., 2001*; *Luong et al., 2013*). It is therefore tempting to speculate that it has esterase activity and targets the LUBAC-assembled chains during its counteraction of

NF-κB activation. However, further work will be necessary to identify potential esterase activity in DUBs including Cezanne.

Understanding the biological functions of the heterotypic chains is a very important aspect. A recent study using HOIL-1L ΔRBR mice and HOIL-1L knock-in cell of ligase inactive mutant showed that HOIL-1L ligase activity regulates TNF-induced signalling and apoptosis (*Fuseya et al., 2020*). In line with their observations, we also observed that HOIL-1L Cys458 mutant as a part of LUBAC increased the linear ubiquitination signal in comparison to the WT control. They also showed that RIPK1, a known substrate of LUBAC, in the TNF complex is linearly ubiquitinated in at increased levels. These data indicate that the HOIL-1L catalytic activity negatively regulates ubiquitination of the substrates in cells, which also supports our observations in vitro. In cells, they could not detect oxyester linkage on HOIL-1L but on HOIP in WT cells, which is abolished in HOIL-1L catalytic inactive cells. Since they detect remaining HOIL-1L mono-ubiquitination signal even when all the Lys residues in HOIL-1L are mutated to Arg, it is probable that there are heterogeneous population of ubiquitination sites in HOIL-1L. Furthermore, this could be due to that oxyester bond being biochemically less stable and being a minor fraction in cells. Since HOIL-1L inactive mutant is no longer ubiquitinated in vitro, upregulated ubiquitination of HOIL-1L mutant in cells (*Fuseya et al., 2020*) may depend on interacting partners existing in cells.

To further understand the roles of these ester-linked branches in biology, it will be important to dissect biochemical characteristics such as their interaction mode with known linear ubiquitin chain-specific binding domains (*Fennell et al., 2018*), existence of DUBs, and other possible ligases that catalyse their formation in future studies.

# Materials and methods

## Plasmids

pGEX-6p1-*Hs*OTULIN, pGEX-6p1-*Hs*HOIP, pGEX-6p1-*Hs*HOIL-1L, and pGEX-6p1-*Hs*SHARPIN (*Fennell et al., 2019*), as well as pGEX-4t1-mNEMO$_{250-412}$ *Wagner et al., 2008* have been previously reported. pGEX-6p1-*Hs*UbcH7 was kindly provided by Katrin Rittinger (Francis Crick Institute, UK). pOPIN-K-*Hs*OTUD1$_{287-481}$ (*Mevissen et al., 2013*), pOPIN-K-*Hs*OTUD2 (*Mevissen et al., 2013*), pOPIN-K-*Hs*OTUD3$_{52-209}$ (*Mevissen et al., 2013*), pOPIN-K-*Hs*Cezanne$_{53-446}$ (*Wagner et al., 2008*), pOPIN-K-*Hs*TRABID$_{245-697}$ (*Licchesi et al., 2011*), pOPIN-K-*Hs*OTUB1 (*Mevissen et al., 2013*), pOPIN-K-vOTU$_{1-183}$ (*Akutsu et al., 2011*), and pOPIN-S-*Hs*Usp21$_{196-565}$ (*Ye et al., 2011*) were kind gifts from David Komander. ORF of ubiquitin T$_{12}$V, T$_{55}$V, and T$_{12/55}$V mutants were generated by PCR and inserted twice into a pETDuet-1 vector by isothermal assembly. pKL-der-*Hs*LUBAC was assembled by inserting *Hs*HOIP, His$_6$-*Hs*HOIL-1L, and Strep(II)-*Hs*Sharpin coding sequences into a pKL-derived vector using BsaI. pKL-der-*Hs*LUBAC-HOIP(C$_{885}$A), pKL-der-*Hs*LUBAC-HOIL-1L (C$_{460}$A), pKL-der-*Hs*LUBAC-HOIP(C$_{885}$A)-HOIL-1L(C$_{460}$A), and pKL-der-*Hs*LUBAC-HOIL-1L(T$_{203}$A/R$_{210}$A) were generated by a standard protocol of site-directed mutagenesis.

## Antibodies

The following antibodies were used in this study: anti-HOIL-1L (clone 2E2; Merck MABC576), anti-HOIP (Merck SAB2102031), anti-IκBα (44D4; Cell Signaling Technology 4812), anti-Ser32/36-phospho-IκBα (5A5; Cell Signaling Technology 9246), anti-SHARPIN (NOVUS Biologicals NBP2-04116), anti-ubiquitin (P4D1; Santa Cruz Biotechnology sc-8017), anti-vinculin (Merck V9131). All antibodies were diluted in TBS 5% (w/v) BSA, 0.05% (v/v) Triton according to the manufacturer's recommended dilutions.

## Generation of anti-linear ubiquitin monoclonal antibody

Mouse monoclonal anti-linear ubiquitin chain antibody (clone LUB4) was generated by immunizing 5-week-old male ICR mice (Charles River Laboratory, Yokohama, Japan) with a neoepitope peptide (LRLRGGMQIFVK) derived from the linear ubiquitin chain, which comprises a ubiquitin C-terminal sequence (amino acids 71–76) and a ubiquitin N-terminal sequence (amino acids 1–6) conjugated to ovalbumin. Cells isolated from the popliteal and inguinal lymph nodes were fused with a mouse myeloma cell line, PAI. Supernatants of the growing hybridomas were tested by direct enzyme-linked

immunosorbent assay and western blot analysis. Specificity of the antibody clone was validated by immunoblotting (*Figure 4—figure supplement 1*).

## Immunoblotting

Protein samples were mixed with SDS -sample buffer containing 5% β-mercaptoethanol and denatured at 96℃ for 5 min. Proteins were separated by SDS-PAGE and transferred to nitrocellulose membranes (GE Healthcare 10600019 or 10600001). Appropriate transfer of proteins was verified by staining of membranes with the Ponceau S solution (Roth 5938.1). Membranes were blocked with TBS 5% BSA (w/v), 0.05% Triton (v/v), and blotted at 4℃ overnight with indicated primary antibodies. Membranes were subsequently blotted with anti-mouse-IgG-HRP conjugate (Bio-Rad 1706516) or anti-rabbit-HRP conjugate (Agilent Dako P044801-2) and visualized with Western Blotting Luminol Reagent (Santa Cruz Biotechnology sc-2048) using Amersham Hyperfilm MP (GE Healthcare) chemiluminescence films.

## Purification of recombinant LUBAC from insect cells

The transfer vector carrying the HsHOIP, His6-HsHOIL-1L, and Strep(II)-HsSHARPIN coding sequences was cloned using the GoldenBac cloning system (*Neuhold et al., 2020*) and transposed into the bacmid backbone carried by DH10EmBacY cells (Geneva Biotech). The bacmid was then used to generate a V0 virus stock in Sf9 cells, which was amplified once to give a V1 virus stock used to infect 1 l cultures of Sf9 cells (Expression Systems). Cells were grown in ESF 921 Insect Cell Culture Medium Protein Free (Expression Systems 96-001-01) at 27℃ and infected at a density of $2 \times 10^6$ cells/ml. Cells were harvested 72 hr after growth arrest and resuspended in 100 mM HEPES, 100 mM NaCl, 100 mM KCl, 100 mM Arg, 10 mM MgSO$_4$, 20 mM imidazole, pH 8.0 supplemented with 1 tablet of cOmplete Mini EDTA-free protease inhibitor cocktail (Merk). Cells were lysed by four passes through a Constant Systems Cell Disruptor at 1.4 kBar and then supplemented with 100 mM Benzonase and 10 mM PMSF. Lysates were cleared by centrifugation at 48,284 g for 45 min at 4℃ and loaded into a HisTrap FF cartridge (GE Healthcare). Proteins were eluted with 100 mM HEPES, 100 mM NaCl, 100 mM KCl, 50 mM arginine, 500 mM imidazole, pH 8.0 and loaded into a Streptactin Superflow cartridge (IBA Lifesciences 2-1238-001). Cartridge was washed with 100 mM HEPES, 100 mM NaCl, 100 mM KCl, pH 8.0, and proteins were eluted with 100 mM HEPES, 100 mM NaCl, 100 mM KCl, 5 mM D-desthiobiotin, pH 8.0. Complexes were concentrated using a Centriprep (Merck 4311) centrifugal filter with a 50 kDa cut-off, then flash-frozen in liquid nitrogen and stored at −80℃.

## Protein purification from *E. coli*

Proteins were expressed in *E. coli* BL21(DE3) cells (Agilent) for 16 hr at 18℃, 25℃, or 30℃. Cell pellets were harvested and lysed by sonication with a Branson 450 Digital Sonifier (Branson) by pulsing for 1.5 min with 1 s pulses and 2 s pauses at amplitude of 60%. Cells expressing GST-NEMO-Strep (II), GST-HOIP, GST-HOIL-1L, GST-SHARPIN, or GST-UbcH7 were lysed in 100 mM HEPES, 500 mM NaCl, pH 7.5 supplemented with 1 tablet of cOmplete Mini EDTA-free protease inhibitor cocktail. Cells expressing ubiquitin mutants were lysed in 50 mM ammonium acetate, pH 4.5 supplemented with 1 tablet of cOmplete Mini EDTA-free protease inhibitor cocktail. Cells expressing His$_6$-SUMO-Usp21$_{196-565}$ were lysed in 50 mM Tris, 200 mM NaCl, pH 8.5 supplemented with 1 tablet of cOmplete Mini EDTA-free protease inhibitor cocktail. Cells expressing GST-OTUD1$_{287-481}$, GST-OTUD2, GST-OTUD3$_{52-209}$, GST-Cezanne$_{53-446}$, GST-TRABID$_{245-697}$, GST-OTUB1, GST-OTULIN, and GST-vOTU$_{1-183}$ were lysed in 50 mM Tris 200 mM NaCl, 5 mM DTT, pH 8.5 supplemented with 1 tablet of cOmplete Mini EDTA-free protease inhibitor cocktail. Cell lysates were supplemented with 500 U Recombinant DNAse (Merck 4716728001) and 100 mM PMSF. For purification of HOIP, HOIL-1L, SHARPIN, NEMO-Strep(II), and UbcH7 lysates were loaded into a GSTrap HP cartridge (GE Healthcare); once loaded GST-PreScission Protease (GE Healthcare) was injected into the column and incubated overnight at 4℃. UbcH7, HOIP, HOIL-1L, and SHARPIN were eluted and further purified over a Superdex 75 16/600 pd or a Superdex 200 16/600 pd column (GE Healthcare) equilibrated in 50 mM HEPES, 150 mM NaCl, pH 7.5. NEMO-Strep(II) was loaded into a Streptactin Superflow cartridge (IBA Lifesciences) and eluted in 50 mM Tris, 150 mM NaCl, 1 mM DTT, 1.5 mM D-desthiobiotin, pH 7.5. For ubiquitin purification, lysates were loaded into a ResourceS (GE Healthcare)

cartridge and eluted in a gradient of 0–500 mM NaCl in 50 mM ammonium acetate, pH 4.5. The protein was further purified over a Superdex 75 16/600 pd column (GE Healthcare) equilibrated in 50 mM HEPES, 150 mM NaCl, pH 7.5. OTUD1$_{287-481}$, OTUD2, OTUD3$_{52-209}$, Cezanne$_{53-446}$, TRABID$_{245-697}$, OTUB1, OTULIN, and vOTU$_{1-183}$ were purified as described in *Hospenthal et al., 2015*, in brief: lysates were loaded into a GSTrap HP cartridge (GE Healthcare) and treated overnight with GST-PreScission Protease (GE Healthcare). Eluted proteins were further purified over a Superdex 75 16/600 pd column (GE Healthcare) equilibrated in 50 mM Tris, 200 mM NaCl, 5 mM DTT, pH 8.5. For purification of Usp21$_{196-565}$, lysate was loaded ino a HisTrap FF cartridge (GE Healthcare) and eluted with 50 mM Tris, 200 mM NaCl, 500 mM imidazole, pH 8.5. Imidazole was removed by buffer exchange using a Vivaspin centrifugal filter (Sartorius) with a 10 kDa cut-off, eluate was treated with SENP2 (R&D Systems E-710–050) overnight, and protein was further purified over a Superdex 75 16/600 pd column (GE Healthcare) equilibrated in 50 mM Tris, 200 mM NaCl, 5 mM DTT, pH 8.5. Proteins were concentrated using Vivaspin centrifugal filters (Sartorius) of appropriate cut-offs, flash-frozen in liquid nitrogen and stored at −80°C. For GST and GST-NEMO$_{250-412}$, they were expressed in BL21 (DE3) cells for 16 hr at 25°C (*Wagner et al., 2008*). Cells were harvested and lysed by sonication in 50 mM Tris, 100 mM EDTA, 50 mM EGTA, 150 mM NaCl, pH 7.5 supplemented with 1 tablet of cOmplete Mini EDTA-free protease inhibitor cocktail, lysates were incubated overnight at 4°C with 400 µl of Glutathione Sepharose 4B per l of expression culture (GE Healthcare GE17-0756-01) with gentle rotation. Beads immobilized samples were washed in 50 mM Tris, 100 mM EDTA, 150 mM NaCl, 0.5% Triton, pH 7.5 and resuspended in 50 mM Tris, pH 7.5 (*Einarson et al., 2007*).

## MP measurement of LUBAC reconstitution

For MP measurements, microscope coverslips (No. 1.5, 24 × 50, VMR) were cleaned by sequential sonication in Milli-Q/isopropanol/Milli Q for 5 min at each step, followed by drying with a clean nitrogen stream. Silicon gasket (3 mm diameter, Grace Bio-labs) to create chambers were also washed with Milli-Q/isopropanol/Milli Q and dried using clean nitrogen stream.

To reconstitute LUBAC from individually purified proteins, namely SHARPIN, HOIP, and HOIL, three proteins were mixed at 400 pmol, each protein was mixed for a 100 µl reaction mixture, and incubated for 10 min at room temperature (RT). Subsequently, the mixture was diluted 100-fold and then repeats were collected for 1 min each, using Refeyn OneMP like set up at the lab of Philipp Kukura. The data was analysed using DiscoverMP software (Refeyn Ltd, Oxford, UK) as described elsewhere. Briefly, six frames were averaged to generate ratiometric frames for analysis and threshold 1 at 0.5 and threshold 2 at 0.25 were used to analyse the ratiometric movie. Contrast of the analysed data was converted to mass using the mass to contrast relation for an in-house mass standard containing protein species at 90, 180, 360, and 540 kDa. Compilation of the mass of the individual repeats and histogram generation was done using in-house python script.

## GST pull-down assays and NH$_2$OH treatment

MEFs were lysed in lysis buffer containing 50 mM HEPES, 150 mM NaCl, 1 mM EDTA, 1 mM EGTA, 1% (v/v) Triton X-100, 10% (v/v) glycerol, 25 mM NAF, 10 µM ZnCl$_2$, 10 mM NEM, 1 mM PMSF, 5 mM Na$_3$VO$_4$, pH 7.4 supplemented with 1 tablet of cOmplete Mini EDTA-free protease inhibitor cocktail. Lysates were cleared by centrifugation at 21,130 g for 15 min at 4°C. Equal amounts of GST or GST-NEMO$_{250-412}$, as determined by SDS-PAGE, were added to lysates and samples were incubated overnight at 4°C with gentle rocking. Beads washed with PBS, then buffer containing 100 mM HEPES, 100 mM NaCl, 100 mM KCl, pH 8.0, then resuspended in a buffer containing 100 mM HEPES, 100 mM NaCl, 100 mM KCl, pH 9.0 ± 1.2 M NH$_2$OH. Reactions were incubated 30 min at 37°C in a Thermomixer comfort (Eppendorf, Germany) thermomixer; reactions were stopped with SDS buffer and incubated at 37°C, then subjected to SDS-PAGE and immunoblotting with the indicated antibodies.

## Negative staining

Recombinant LUBAC was run through a Superdex S200 Increase 3.2/300 column equilibrated in 100 mM HEPES, 100 mM NaCl, 100 mM KCl, pH 8.0. Fractions containing the peaks for recombinant LUBAC were collected for staining on 400-mesh Cu/Pd grids (Agar Scientific G2440PF) coated with a 3-nm-thick continuous carbon support film. For staining grids were glow discharged for 1 min at

200 mA and $10^{-1}$ mBar in a BAL-TEC SCD005 sputter coater (BAL-TEC, Liechtenstein), samples were pipetted onto the grid and incubated for 1 min before blotting. Grids were then stained with 2% (w/v) uranyl acetate, pH 4 (Merck) for 1 min, blotted dry, and left to air-dry for 10 min at RT and pressure before being stored in a desiccator.

## Electron microscopy

Grids were screened on a FEI Morgani 268D transmission electron microscope (Thermo Fisher Scientific, Amsterdam, The Netherlands) operated at 80 kV using a 300 µm condenser lens aperture and 50 µm objective lens aperture; images were recorded on an 11 megapixel Morada CCD camera (Olympus-SIS, Germany). For data acquisition grids were imaged on a FEI Tecnai G2 20 (Thermo Fisher Scientific, Amsterdam, The Netherlands) operated at 200 kV using a 200 µm condenser lens aperture and 100 µm objective lens aperture; imaged on a FEI Eagle 4 k HS CCD camera (Thermo Fisher Scientific, Amsterdam, The Netherlands) with a pixel size of 1.85 Å/pixel.

## Image processing of negative stain data

The negative stain data was processed in relion 3.1 (*Zivanov et al., 2018*; *Scheres, 2012a*; *Scheres, 2012b*) unless otherwise stated. CTF parameters were determined using CTFFIND4 (*Rohou and Grigorieff, 2015*). Micrographs that suffered from drift were excluded from the further analysis; 600 particles were picked manually and classified in 2D to yield templates for autopicking. Autopicking yielded 637,000 particles that were extracted in a box of 180 pixels but cropped in Fourier space to 90 pixels. The dataset was split in six subsets that were subjected to 2D classification. Class averages showing distinct particle features were kept and their particles were combined for a final round of 2D classification yielding 163,000 remaining particles. These particles were subjected to initial model generation yielding three models that are very similar in shape. The most isotropic model was used for 3D classification once again yielding similar models. All steps were performed without CTF correction to avoid artifacts and over-fitting. Back-projections of the final map were generated to validate the self-consistency of the model.

## Cross-linking mass spectrometry

For cross-linking preparations, 100 µg of recombinant LUBAC were cross-linked in 100 mM HEPES, 100 mM NaCl, 100 mM KCl, pH 8.0 with 3.75 mM DMTMM (Merck 74104) for 1 hr at RT. The reaction cross-linker was removed using a Zeba Micro Spin desalting column (Thermo Fisher Scientific 89883). Samples were dried at 45°C in a vacuum centrifuge connected to a cooling trap (Heto Lab Equipment) and then resuspended in 8 M urea. Proteins were subjected to reduction with 2.5 mM TCEP and subsequently alkylated with 5 mM iodoacetamide for 30 min at RT after which samples were diluted 1:7 (v/v) with 50 mM ammonium bicarbonate (ABC). Samples were digested with 2 µg Trypsin (Promega V5280) for 20 hr at 37°C, samples were then digested for a further 3 hr with 2 µg Chymotrypsin (Roche 11418467001) at 25°C. Digest was quenched with 0.4% (v/v) trifluoroacetic acid (TFA) and samples were loaded on a Sep-Pak C18 cartridge (Waters WAT054955) equilibrated with 5% (v/v) acetonitrile, 0.1% (v/v) formic acid (FA); samples were eluted with 50% (v/v) acetonitrile, 0.1% (v/v) FA. Cross-linked peptides were enriched in a Superdex 30 Increase 3.2/300 column (GE Healthcare) equilibrated with 30% (v/v) acetonitrile, 0.1% (v/v) TFA. Fractions containing cross-linked peptides were evaporated to dryness and resuspended in 5% (v/v) acetonitrile, 0.1% (v/v) TFA. Cross-linked peptides were separated on an UltiMate 3000 RSLC nano HPLC system (Thermo Fisher Scientific) coupled to an Orbitrap Fusion Lumos Tribid mass spectrometer (Thermo Fisher Scientific) equipped with a Proxeon nanospray source (Thermo Fisher Scientific). Peptides were loaded onto an Acclaim PepMap 100 C18 HPLC column (Thermo Fisher Scientific 160454) in a 0.1% TFA mobile phase. Peptides were eluted into an Acclaim PepMap 100 C18 HPLC column (Thermo Fisher Scientific 164739) in a binary gradient between mobile phase A (99.9/0.1% v/v water/FA) and mobile phase B (19.92/80/0.08% v/v/v water/acetonitrile/FA). The gradient was run from 2% to 45% mobile phase B over 3 hr and then increased to 90% B over 5 min.

Measurement was performed in data-dependent mode with 3 s cycle time at a resolution of 60,000 in the m/z range of 350–1500. Precursor ions with a charge state of +3 to +7 were fragmented by HCD with collision energy of 29% and fragments were recorded with a resolution of 45,000 and precursor isolation width of 1.0 m/z. A dynamic exclusion time of 30 s was used.

Fragment spectra peak lists were generated with MSConvert v3.0.9974 (*Chambers et al., 2012*) with a peak-picking filter. Cross-link identification was done using XiSearch v1.6.742 (*Giese et al., 2016*) with 6 ppm $MS^1$-accuracy, 20 ppm $MS^2$-accuracy, DMTMM cross-linking was set with asymmetric reaction specificity for lysine and arginine to glutamate and aspartate; carbamidomethylation of cysteine was a fixed modification, oxidation of methionine was a variable modification, trypsin/chymotrypsin digest was set with up to four missed cleavages, and all other variables were left at default settings. Identified cross-links were filtered to 10% FDR at the link level with XiFDR v1.1.27 (*Fischer and Rappsilber, 2017*), links with FDR < 5% were visualized with xVis (*Grimm et al., 2015*).

## In vitro ubiquitination assay

Reaction mixtures were prepared containing Ube1, UbcH7, and either LUBAC or HOIP, HOIL-1L, and SHARPIN with all proteins at 0.338 µM final concentration in a buffer containing 50 mM HEPES, 150 mM NaCl, 0.5 mM $MgSO_4$, pH 8.0 (described in *Ikeda et al., 2011*). Reactions were started by addition of 59.0 µM ubiquitin or fluorescein-tagged ubiquitin, 2 mM ATP, and 0.5 µg of NEMO where indicated, then incubated at 37°C in a Mastercycler nexus (Eppendorf, Germany) thermocycler for 2 hr unless otherwise stated. Reaction was stopped by addition of SDS buffer and boiling at 95°C, proteins were resolved by SDS-PAGE, and analysed by immunoblotting with the indicated antibodies or by fluorescence imaging. Recombinant human ubiquitin, $His_6$-ubiquitin, and ubiquitin KR mutants ($K_6R$, $K_{11}R$, $K_{27}R$, $K_{29}R$, $K_{33}R$, $K_{48}R$, $K_{63}R$, $K_0$) were purchased from Boston Biochem.

## $NH_2OH$ treatment of in vitro assembled poly-ubiquitin chains

LUBAC in vitro and UbiCRest reactions were mixed with 1.2 M $NH_2OH$ in 40 mM HEPES, 40 mM NaCl, 40 mM KCl, pH 9.0, and reactions were incubated at 37°C in a Thermomixer comfort (Eppendorf, Germany) thermomixer for 2 hr. Reactions were stopped by addition of SDS buffer and incubated at 37°C, then subjected to SDS-PAGE and immunoblotting with the indicated antibodies.

## UbiCRest assay

Protocol was adapted from *Hospenthal et al., 2015*; in brief, DUBs were diluted to 0.8 µM ($OTUD1_{287-481}$, OTUD2, $OTUD3_{52-209}$, $Cezanne_{53-446}$, $TRABID_{245-697}$, OTUB1, and OTULIN) or 10 µM ($vOTU_{1-183}$, $Usp21_{196-565}$) in 25 mM Tris, 150 mM NaCl, 10 mM DTT, pH 7.5, and incubated at RT for 10 min. One LUBAC in vitro chain assembly reaction was divided into aliquots and mixed at a 1:1 (v/v) ratio with each DUB in 50 mM Tris, 50 mM NaCl, 5 mM DTT, pH 7.5, and reactions were incubated at 37°C in a Thermomixer comfort (Eppendorf, Germany) thermomixer for 2 hr. Reactions were stopped as described above and analysed by immunoblotting with the indicated antibodies.

## Mass spectrometry

LUBAC in vitro chain assembly reactions were resolved on 4–15% Mini-PROTEAN TGX gels (Bio-Rad 4561083) and stained with Instant Blue (Merck ISB1L). Fragments were excised from the gel and slices were washed sequentially with 100 mM ABC, then 50% (v/v) acetonitrile 50 mM ABC at 57°C for 30 min. Gel slices were shrunk in 100% acetonitrile and reduced with 1 mg/ml DTT in 100 mM ABC. Slices were then alkylated with MMTS in 100 mM ABC at RT for 30 min. Gel slices were subjected to tryptic digest (Promega V5280) overnight at 37°C. Peptides were extracted from gel slices by sonication in 5% (v/v) FA. Peptides were loaded into a PepMap C18 (5 mm × 300 µm ID, 5 µm particles, 100 Å pore size; Thermo Fisher Scientific) trap column on an UltiMate 3000 RSLC nano HPLC system (Thermo Fisher Scientific, Amsterdam, The Netherlands) coupled to a Q Exactive HF-X mass spectrometer (Thermo Fisher Scientific, Bremen, Germany) equipped with a Proxeon nanospray source (Thermo Fisher Scientific, Odense, Denmark) using a solution of 0.1% (v/v) TFA as the mobile phase. Samples were loaded at a flow rate of 25 µl/min for 10 min and then eluted into an analytical C18 (500 mm × 75 µm ID, 2 µm, 100 Å; Thermo Fisher Scientific, Amsterdam, The Netherlands) column in a binary gradient between mobile phase A (99.9/0.1% v/v water/FA) and mobile phase B (19.92/80/0.08% v/v/v water/acetonitrile/FA). The gradient was run from 98%/2% A/B to 35%/65% A/B over 1 hr; the gradient was then adjusted to 5%/95% A/B over 5 min and held for a further 5 min before returning to 98%/2% A/B. The Q Exactive HF-X mass spectrometer was operated in data-dependent mode, using a full scan (m/z range 380–1500, nominal resolution of 60,000, target value $1_E$; *Wang et al., 2007*) followed by MS/MS scans of the 10 most abundant ions. MS/MS

spectra were acquired using normalized collision energy of 28, isolation width of 1.0 m/z, resolution of 30,000, and target value was set to $1_E$ (*Vosper et al., 2009*). Precursor ions selected for fragmentation were put on a dynamic exclusion list for 20 s (excluding charge states 1, 7, 8, >8). The minimum AGC target was set to $5_E$ (*McClellan et al., 2019*) and intensity threshold was calculated to be $4.8_E$ (*McDowell and Philpott, 2013*); the peptide match feature was set to preferred and the exclude isotopes feature was enabled. For peptide identification, the generated .raw files were loaded into Proteome Discoverer (version 2.3.0.523, Thermo Fisher Scientific). All hereby created MS/MS spectra were searched using MSAmanda v2.0.0. 12368 (*Dorfer et al., 2014*). The .raw files were searched against the *Spodoptera frugiperda* database (23,492 sequences; 12,713,298 residues) and the swissprot-ecoli database (4418 sequences; 1,386,900 residues) to generate a .fasta file; search parameters used were peptide mass tolerance ± 5 ppm, fragment mass tolerance 15 ppm, maximum of two missed cleavages; the result was filtered to 1% FDR on protein level using Percolator algorithm integrated in Thermo Proteome Discoverer. The sub-database was generated for further processing. The .raw files were then searched using the generated .fasta file using the following search parameters: β-methylthiolation on cysteine was set as a fixed modification; oxidation of methionine, deamidation of asparagine and glutamine, acetylation of lysine, phosphorylation of serine, threonine and tyrosine; methylation of lysine and arginine, di-methylation of lysine and arginine, tri-methylation of lysine, biotinylation of lysine, carbamylation of lysine, ubiquitination of lysine, serine, threonine, and tyrosine were set as variable modifications. Monoisotopic masses were searched within unrestricted protein masses for tryptic enzymatic specificity. The peptide mass tolerance was set to ±5 ppm, fragment mass tolerance to ±15 ppm, maximum number of missed cleavages was set to 2, results were filtered to 1% FDR on protein level using Percolator algorithm (*Käll et al., 2007*) as integrated in Proteome Discoverer. The localization of the posttranslational modification sites within the peptides was performed with the tool ptmRS, based on the tool phosphoRS (*Taus et al., 2011*). Peptide areas were quantified using in-house-developed tool apQuant (*Doblmann et al., 2019*).

## Generation of C57BL/6J *Rbck1*$^{C458A/C458A}$ mice and genotyping

C57BL/6J *Rbck1*$^{C458A/C458A}$ mice were generated as described elsewhere (*Fennell et al., 2019*). The gRNA targeting sequence was designed using the online tool ChopChop (http://chopchop.cbu.uib.no), two separate gRNA sequences were selected (gRNA1 and gRNA63). The gRNA sequences were inserted to pX330-U6-Chimeric_BB-CBh-hSpCas9 (a gift from Feng Zhang, Addgene plasmid #42230) (*Cong et al., 2013*) as annealed oligonucleotides (*Supplementary file 4*) using BbsI. A T7-gRNA product was amplified by PCR and used for in vitro transcription using the MEGAshortscript T7 kit (Invitrogen AM1345). The in vitro transcribed gRNA was purified with the MEGAclear kit (Invitrogen AM1908). A single-stranded donor template oligonucleotide (ssOligo) was designed containing the C458A mutation, silent mutation of the PAM site, and a silent Hpy188III restriction site (5′ CGGATTGTGGTCCAGAAGAAAGACGGCGCTGACTGGATTCGCTGTACAGTCTGCCACACTGAGA TC 3′). Superovulation was induced in 3- to 5-week-old female C57BL/6J donor mice by treatment with 5IU of pregnant mare's serum gonadotropin (Hölzel Diagnostika OPPA01037) and 5IU of human chorionic gonadotropin (Intervet, GesmbH); females were mated and zygotes were isolated in M2 media (Merck Milipore MR-015P-D). Zygotes were cultured in KSOM medium (Cosmo Bio Co., Ltd RB074) and cytosolically injected with 100 ng/µl Cas9 mRNA (Merck CAS9MRNA-1EA), 50 ng/µl gRNA, and 200 ng/µl ssOligo; injected zygotes were transferred to pseudo-pregnant females. Correct knock-in was confirmed in founder mice by Sanger sequencing. Routine genotyping of mice was done by digestion with Hpy188III (NEB R0622) and analysis on a 2% agarose gel.

## Mouse husbandry

C57BL/6J *Rbck1*$^{C458A/C458A}$ mice were bred and kept at the IMBA/IMP animal house. All mice were bred and maintained in accordance with ethical animal license protocols complying with the Austrian and European legislation. Animal procedures were covered by the license 568809/2013/18.

## Isolation and immortalization of MEFs

MEF isolation and immortalization methods are described elsewhere (*Fennell et al., 2019*). Briefly, primary MEFs were isolated at E13.5 following standard protocols. Pregnant females were sacrificed,

and the uterus was extracted. Isolated tissue from the embryo was digested with trypsin (Thermo Fisher 25300054) for 5 min at 37℃ and digest was quenched with Dulbecco's modified eagle's medium (DMEM) containing 10% (v/v) FCS. Cells were collected by centrifugation and cultured at 37℃, 5% $CO_2$ in DMEM (Sigma D5648) supplemented with 10% (v/v) FCS, 0.2 U/ml penicillin, 0.2 μg/ml streptomycin (Merck P0781), and 2 mM L-glutamine (Gibco 25030–024). MEFs were immortalized with SV40 large T-antigen by transfection using GeneJuice transfection reagent (Merck Millipore 70967). Our routine mycoplasma tests showed negative results for these cells.

## TNF stimulation of MEFs
$1.6 \times 106$ MEFs were seeded in a 10 cm cell culture dish and grown overnight, serum-starved overnight in DMEM supplemented with 0.2% (v/v) FCS, 0.2 U/ml penicillin, 0.2 μg/ml streptomycin, and 2 mM L-glutamine. The following day cells were stimulated for 15 min with 50 ng/ml mouse TNF (Peprotech).

## Acknowledgements
We thank Dr Gijs Versteeg (Max F Perutz Laboratories, Austria) and Katrin Rittinger (The Crick Institute, UK) for critical discussion and feedback on the work and manuscript. We thank Mayu Seida (MIB, Kyushu University, Japan), Olga Olszanska, Steven Dupart, and Lilian Fennell (IMBA, Austria) for their technical support. We also thank Richard Imre, Elisabeth Roitinger, and Otto Hudecz for mass spectrometry data analysis, as well as Ines Steinmacher and Susanne Opravil for technical support (the Protein Chemistry Facility, IMP/IMBA core facility, Austria). Samples were prepared and data was recorded at the EM Facility of the Vienna BioCenter Core Facilities GmbH (VBCF, Austria). We thank the IMP/IMBA core facilities, including Transgenic Service, Comparative Medicine, and Molecular Biology Service for their technical support. Baculovirus production and insect cell culture was performed by the Protein Technologies Facility (VBCF, Austria). Research at the Ikeda Lab is supported by JSPS KAKENHI (Grant Number JP 18K19959, JP 21H04777, JP 21H00288) and the Austrian Academy of Sciences. Research at the Haselbach Lab is supported by Boehringer Ingelheim. Research at the Clausen Lab is supported by the FFG Headquarter Grant 852936 and Boehringer Ingelheim. Research at the Kukura Lab is supported by the ERC Starting Grant PHOTOMASS 819593.

## Additional information

### Competing interests
Manish S Kushwah: a consultant in Refeyn Ltd. Shinji Sakamoto: an employee of JT Inc. Philipp Kukura: academic founder, consultant, and shareholder in Refeyn Ltd. The other authors declare that no competing interests exist.

### Funding

| Funder | Grant reference number | Author |
|---|---|---|
| Japan Society for the Promotion of Science | JP 18K19959 | Fumiyo Ikeda |
| Japan Society for the Promotion of Science | JP 21H04777 | Fumiyo Ikeda |
| Japan Society for the Promotion of Science | JP 21H00288 | Fumiyo Ikeda |
| Austrian Academy of Sciences | | Fumiyo Ikeda |
| Boehringer Ingelheim | | Tim Clausen David Haselbach |
| FFG | 852936 | Tim Clausen |
| European Research Council | PHOTOMASS 819593 | Philipp Kukura |

The funders had no role in study design, data collection and interpretation, or the decision to submit the work for publication.

## Author contributions

Alan Rodriguez Carvajal, Data curation, Formal analysis, Validation, Investigation, Visualization, Methodology, Writing - original draft, Writing - review and editing; Irina Grishkovskaya, Manish S Kushwah, Data curation, Validation, Visualization; Carlos Gomez Diaz, Resources, Data curation, Methodology; Antonia Vogel, Data curation, Formal analysis, Visualization; Adar Sonn-Segev, Data curation, Formal analysis, Investigation; Katrin Schodl, Luiza Deszcz, Zsuzsanna Orban-Nemeth, Investigation, Methodology; Shinji Sakamoto, Resources; Karl Mechtler, Resources, Methodology; Philipp Kukura, Data curation, Formal analysis, Supervision, Funding acquisition, Methodology; Tim Clausen, Formal analysis, Supervision, Funding acquisition, Methodology; David Haselbach, Conceptualization, Resources, Data curation, Software, Formal analysis, Supervision, Funding acquisition, Investigation, Methodology, Writing - review and editing; Fumiyo Ikeda, Conceptualization, Supervision, Funding acquisition, Investigation, Visualization, Writing - original draft, Project administration, Writing - review and editing

## Author ORCIDs

Alan Rodriguez Carvajal (iD) https://orcid.org/0000-0002-8340-8620
Carlos Gomez Diaz (iD) http://orcid.org/0000-0002-6416-806X
Shinji Sakamoto (iD) https://orcid.org/0000-0003-2480-2940
Tim Clausen (iD) http://orcid.org/0000-0003-1582-6924
David Haselbach (iD) https://orcid.org/0000-0002-5276-5633
Fumiyo Ikeda (iD) https://orcid.org/0000-0003-0407-2768

## Ethics

Animal experimentation: All mice were bred and maintained in accordance with ethical animal license protocols complying with the Austrian and European legislation. Animal procedures were covered by the license 568809/2013/18.

## Decision letter and Author response

Decision letter https://doi.org/10.7554/eLife.60660.sa1
Author response https://doi.org/10.7554/eLife.60660.sa2

# Additional files

## Supplementary files

• Supplementary file 1. Population masses determined by mass photometry.

• Supplementary file 2. Unique inter-protein cross-links detected by linear ubiquitin chain assembly complex (LUBAC) 4-(4,6-dimethoxy-1,3,5-triazin-2-yl)-4-methylmorpholinium tetrafluoroborate (DMTMM ) XL-MS analysis.

• Supplementary file 3. Unique intra-protein cross-links detected by linear ubiquitin chain assembly complex (LUBAC) 4-(4,6-dimethoxy-1,3,5-triazin-2-yl)-4-methylmorpholinium tetrafluoroborate (DMTMM) XL-MS analysis.

• Supplementary file 4. Primers.

• Transparent reporting form

## Data availability

All data besides the structural data and the mass spec data generated or analysed during this study are included in the manuscript and supporting files. Source data files have been provided for Supplementary Tables 1-3.

The following datasets were generated:

| Author(s) | Year | Dataset title | Dataset URL | Database and Identifier |
|---|---|---|---|---|
| Ikeda F, Rodriguez Carvajal A, Haselbach D | 2021 | Linear Ubiquitin Chain Assembly Complex | https://www.ebi.ac.uk/pdbe/entry/emdb/EMD-11054 | Electron Microscopy Data Bank, EMD-11054 |
| Ikeda F, Vogel A | 2021 | The linear ubiquitin chain assembly complex LUBAC generates heterotypic ubiquitin chains | https://www.ebi.ac.uk/pride/archive/projects/PXD019771 | PRIDE, PXD019771 |

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
