## [Decision Letter]

**Acceptance summary:**

This work provides important new insights into the architecture of LUBAC and the activity of its HOIL-1L E3 ligase component in synthesizing heterotypic, branched ubiquitin chains through oxyester-bond formation. The spatial proximity of HOIL-1L's catalytic domain and the catalytic domain of HOIP, another E3 ligase within LUBAC, indicates an intriguing ubiquitin relay mechanism that may be prevalent among E3 ligases.

**Decision letter after peer review:**

Thank you for submitting your article "The linear ubiquitin chain assembly complex LUBAC generates heterotypic ubiquitin chains" for consideration by *eLife*. Your article has been reviewed by 3 peer reviewers, and the evaluation has been overseen by a Reviewing Editor and Cynthia Wolberger as the Senior Editor. The following individual involved in review of your submission has agreed to reveal their identity: Gabe Lander (Scripps) (Reviewer #1).

The reviewers have discussed the reviews with one another and the Reviewing Editor has drafted this decision to help you prepare a revised submission.

The editors have judged that your manuscript is of significant interest, yet some additional experiments are required before publication, as described. We would like to draw your attention to changes in our revision policy that we have made in response to COVID-19 (https://elifesciences.org/articles/57162). Because many researchers have temporarily lost access to the labs, we will give authors as much time as they need to submit revised manuscripts.

Summary:

This manuscript by Carvajal et al. provides novel insights into HOIL-1L's activity within the LUBAC complex in synthesizing heterotypic, branched ubiquitin chains through oxyester-bond formation. The authors successfully produced and isolated recombinant LUBAC, containing full length HOIL-1L, HOIP, and SHARPIN, and although the intrinsic flexibility prevented a higher-resolution 3D-structure determination, negative-stain EM combined with crosslinking mass spec revealed important new information about the architecture of this complex. Based on the observed spatial proximity of HOIL-1L's and HOIP's catalytic RBR domains, the authors propose an intriguing ubiquitin relay mechanism between these E3 ligases in LUBAC.

The reviewers agreed that this work represents an important contribution to the field, as it corroborates and extends previous findings of HOIL-1L's non-lysine esterification activity. However, the advance and impact could be improved by some additional experiments to further strengthen the mechanistic conclusions.

Essential revisions:

1) Although the authors present the first 3D reconstruction of LUBAC, the contribution of this low resolution model to this study seems questionable. The 2D analyses demonstrate the substantial flexibility of the complex, and projections generated from the 3D structure only marginally match the selected projections shown in Figure 2. If EM analyses are meant to support the biochemical reconstitution of the active LUBAC complex, then the 2D class averages are more than sufficient. Based on the 2D data and the fact that there are many class averages that are not recapitulated by 2D projections (and vice versa), it is highly unlikely that the purified complex is consistent with a single 3D structure. If the authors were able to use negative stain of complexes, where individual subunits contained identifiable tags (e.g. GFP, MBP), to localize subunits and corroborate the crosslink-MS, perhaps a 3D model would be appropriate, but as it stands, the utility of the 3D density seems moot.

2) The circus plot in Figure 3B to represent the crosslinking-MS data could be improved by "weighing" the quantity or confidence of observed crosslinks, such that attention is drawn to the most important and obvious linkages, for instance by using different line widths, color shades, or the presentation of multiple plots at distinct cutoff values.

Furthermore, the pair-wise domain representation in Figure 3C-E gives the impression that a single domain (or even single residue) is found crosslinking to almost every part of the opposing protein (a straight line in the plot which contains many dots) in several instances. This could similarly benefit from thresholding or a more cautious description. Can it truly be inferred that the RBR's and NZF's of HOIP and HOIL-1L are forming a catalytic center, when grey linker-regions are over-represented in the plot?

The authors should discuss the promiscuous crosslinking of linker regions and whether this may indicate high dynamics that could be related to the difficulties in solving better-resolution structures. Also, it may also be visually more appealing to represent non-domain grey regions by a thinner line than known domains in all representations of Figures 3A-3E and 6D.

Based on their crosslinking results, the authors emphasize the interaction of the RBR domains of HOIP and HOIL-1L and speculate that LUBAC may have a single catalytic center. However, since multiple contacts between LUBAC domains are detected (Figures 3B-E), the authors need to explain why they focused on this particular one. It will be interesting to analyze the effect of E2 or E2~Ub on crosslinking.

3) Thr12 and Thr55 were identified as potential ester linkage sites within poly-Ub species, but their mutation did not abolish the formation of hydroxylamine-sensitive bands. The authors should state the observed ubiquitin-sequence coverage in their mass spec experiment and which regions were not covered.

4) The authors hypothesize that a key function of the HOIL-1 esterification activity is to form heterotypic chains. While this might be the case, the alternative hypothesis of HOIL-1 priming substrates with ester-linked ubiquitins that are then linearly extended by HOIP seems also equally valid, especially because multiple substrates have been reported to be modified with linear chains, but HOIP appears to be tailored to modify only an ubiquitin substrate. The authors should discuss this alternative hypothesis and why both systems might be important.

In further support of substrates potentially being the most abundant ester-linked species, NEMO-enriched linear chains from TNF treated cells show a much more pronounced collapse compared to the ester-linked Ub-Ub linkages produced in the absence of substrate in vitro. It would greatly strengthen the paper if the authors could add a recombinant substrate (e.g. IRAK1/2 or MyD88) to the in vitro reaction.

5) In Figure 4B, why could the mixed LUBAC subunits generate a linear chain, but not an oxyester-linked branched Ub4? Do they form a high molecular weight complex that can be detected by gel filtration?

6) In Figures 4E and 5A, it is interesting that Cezanne and vOTU could cleave ester-linked branched Ub4, although the molecular bases for these reactions remain unclear. Are the LUBAC-generated, hydroxylamine-sensitive His-Ub3 and Ub2 shown in Figure 5B cleavable by Cezanne and vOTU?

7) To confirm that the residual oligomeric Ub species after OTULIN treatment (Figure 4E) are exclusively ester-linked, a subsequent hydroxylamine treatment step should be performed.

8) Finally, the suggestion that HOIP-HOIL Ub relay might be at play is exciting and implies that E3-mediated Ub relay might be a prevalent process. In principal it should be possible to test this by impairing E2 binding to the RING1 domain in HOIL in LUBAC. A steric mutation (e.g. X to Arg) would be a more elegant approach than a zinc-coordinating cysteine mutation. If relay is at play, then such mutated LUBAC should still be able to form ester linkages.

---

## [Author Response]

Essential revisions:1) Although the authors present the first 3D reconstruction of LUBAC, the contribution of this low resolution model to this study seems questionable. The 2D analyses demonstrate the substantial flexibility of the complex, and projections generated from the 3D structure only marginally match the selected projections shown in Figure 2. If EM analyses are meant to support the biochemical reconstitution of the active LUBAC complex, then the 2D class averages are more than sufficient. Based on the 2D data and the fact that there are many class averages that are not recapitulated by 2D projections (and vice versa), it is highly unlikely that the purified complex is consistent with a single 3D structure.

We agree with the reviewers that the 2D class averages indicate a degree of flexibility in the complex. The 2D class averages show an asymmetric crescent structure where the mass of the complex is focused at either end with a thinner linker region that is expected to be the source of potential structural flexibility. As such, the 3D model in our first submission may indeed represent an average of a few conformers that differ by the degree of pivoting around the central linker region. However, the 2D class averages are not suggestive of multiple highly variable structural configurations in the complex that would jeopardize the accuracy of the model at the low resolution of negative staining. Nevertheless, to address the reviewers’ concern, we further refined the 3D model of the complex and arrived at a reconstruction that more accurately reflects the 2D class averages (Figure 2B new). Moreover, the projections obtained from the new model match the 2D class averages of the complex much better than those originally presented (Figure 2C new). Therefore, we present a new 3D refined model as a more accurate reconstruction of the LUBAC complex.

If the authors were able to use negative stain of complexes, where individual subunits contained identifiable tags (e.g. GFP, MBP), to localize subunits and corroborate the crosslink-MS, perhaps a 3D model would be appropriate, but as it stands, the utility of the 3D density seems moot.

We certainly agree with the reviewers that identification of the catalytic C-termini of HOIL-1L and HOIP in the structure would corroborate the crosslinking-MS data and increase the usefulness of the structure. We therefore tagged the HOIP or HOIL-1L with an identifiable C-terminal GFP tag to localize their respective C-terminal regions in the structure. The GFP-tagged complexes could be purified and the purity of the preparations could be determined by mass photometry measurements showing the precise expected mass shift from the GFP tag as shown in Author response image 1.

**Author response image 1. sa2fig1:** GFP-tagged LUBAC purification. (A) Purification of LUBAC containing GFP-tagged HOIP or HOIL-1L. (B) Mass photometry measurement of GFP-tagged LUBAC showing 27 kDa shift in the monomeric (250kDa) and dimeric (500kDa) mass of the complex.

Unfortunately, despite the quality of the purifications, we could not reliably identify the GFP tag in the obtained 2D class averages or 3D reconstructions as shown in Author response image 2.

**Author response image 2. sa2fig2:** GFP-tagged LUBAC class averages. (A) Class averages of GFP-tagged LUBAC. (B) Selected class averages that may show GFP tag.

We could not determine the exact reason for the absence of the tag from the negative stain data but we believe it could be the result of either the tag being too flexible with respect to LUBAC for visualization or because the C-termini of the proteins may not be resolved in our model. We appreciate the limitations of our presented model; nevertheless we have included the 3D reconstruction as it represents the first 3D model of the LUBAC complex, which is an important advancement in the field.

2) The circus plot in Figure 3B to represent the crosslinking-MS data could be improved by "weighing" the quantity or confidence of observed crosslinks, such that attention is drawn to the most important and obvious linkages, for instance by using different line widths, color shades, or the presentation of multiple plots at distinct cutoff values.

A score for each crosslink is generated by the software employed. However, multiple cross-linked peptides were detected for all plotted crosslinks. Hence, across all three datasets there are multiple scores for all detected crosslinks falling within the same range. It would therefore not be possible to select scores in an unbiased manner for the plot nor to make a clear visual distinction in the figure given that the scores lie in similar ranges for all the plotted crosslinks.

Furthermore, the pair-wise domain representation in Figure 3C-E gives the impression that a single domain (or even single residue) is found crosslinking to almost every part of the opposing protein (a straight line in the plot which contains many dots) in several instances. This could similarly benefit from thresholding or a more cautious description. Can it truly be inferred that the RBR's and NZF's of HOIP and HOIL-1L are forming a catalytic center, when grey linker-regions are over-represented in the plot?The authors should discuss the promiscuous crosslinking of linker regions and whether this may indicate high dynamics that could be related to the difficulties in solving better-resolution structures.

As the reviewers raise an important point we tried to improve our data by further purifying LUBAC with a gel filtration step prior to crosslinking. Under the new conditions we could not observe any appreciable change in the crosslinking pattern as exemplified in the pair-wise plot for HOIP-to-HOIL-1L inter-protein crosslinks (Author response image 3).

**Author response image 3. sa2fig3:** XL-MS results with additional purification step prior to crosslinking.

Each of the three proteins had one residue that was observed to crosslink indiscriminately to all other domains giving rise to the observed pattern on the pair-wise representations. These residues were HOIL-1L K174, SHARPIN K318, and HOIP K454/458. We argue that these residues are likely found on flexible regions of the proteins thus giving rise to the high levels of crosslinking to all domains in the complex. Indeed, foldindex prediction of HOIL-1L shows that unfolded regions match with these highly crossed-linked regions (Author response image 4). This issue is likely further exacerbated by the dimerization of LUBAC. Therefore, we do not expect all the crosslinks formed by these residues to be structurally relevant. However, as they are a product of the experimental set up they cannot be filtered out during data analysis. Hence, they were left in the figures in order to present an unbiased picture of the data. Nevertheless, the accuracy of the selectively crosslinking regions, including the NZF and RBR domains of HOIL-1L and HOIP, is independent to the background generated by promiscuously crosslinking residues. We are confident in our inference over the catalytic center of LUBAC, as it is supported by our biochemical data. As shown in figure 7, mutation of the catalytic Cys of HOIP inhibits the catalytic activity of HOIL-1L; similarly mutation of the NZF of HOIL-1L abrogates its catalytic activity.

**Author response image 4. sa2fig4:** Fold-index plot of human HOIL-1L. Regions in HOIL-1L observed to cross-link mostly reside within the low fold-index (unfolded) area in the shown plot.

We appreciate that our discussion pertaining to these promiscuous crosslinks may not have been sufficient and have updated the text to reflect the reviewers’ concerns as follows:

“We detected some highly crosslinking residues that formed crosslinks indiscriminately to all subunits of all proteins. […] The presence of flexible regions on the three LUBAC components may be related to the difficulties in determining the structures of full-length HOIP, HOIL-1L, and SHARPIN thus far.”

Also, it may also be visually more appealing to represent non-domain grey regions by a thinner line than known domains in all representations of Figures 3A-3E and 6D.Based on their crosslinking results, the authors emphasize the interaction of the RBR domains of HOIP and HOIL-1L and speculate that LUBAC may have a single catalytic center. However, since multiple contacts between LUBAC domains are detected (Figures 3B-E), the authors need to explain why they focused on this particular one. It will be interesting to analyze the effect of E2 or E2~Ub on crosslinking.

We thank the reviewers for their observation. We also thank the reviewers for interesting suggestions on E2 and Ub-loaded E2 to include in the XL-MS analysis. Due to high complexity of the cross links as described as above, we decided not to include E2 in the new experiments. To explain why we focused on these crosslinks, we have updated the text as follows:

“Interestingly, we observed crosslinks between the HOIL-1L RING1 and HOIP RING1/LDD domains, as well as between the HOIL-1L RING2 and the HOIP RING1/IBR/RING2/LDD domains, which could indicate that the two enzymes have spatially connected catalytic activities. Additionally, HOIL-1L intra-protein crosslinks were formed between its NZF domain and its RING1/IBR/RING2 domains, which could implicate the HOIL-1L NZF domain of unknown function in the catalytic action of HOIL-1L.”

3) Thr12 and Thr55 were identified as potential ester linkage sites within poly-Ub species, but their mutation did not abolish the formation of hydroxylamine-sensitive bands. The authors should state the observed ubiquitin-sequence coverage in their mass spec experiment and which regions were not covered.

We had almost full sequence coverage of ubiquitin in the mass spectrometry samples where Thr 12 and Thr55 were identified as ester-linkage sites. Only the final three amino acids of ubiquitin were not detected in their un-conjugated form, which is expected due to the high sensitivity of the C-terminal LRLRGG peptide to tryptic digest. We have updated the text to reflect the coverage of the ubiquitin sequence as follows:

“MS/MS spectra of GG-conjugated dipeptides at residues Thr12 and Thr55 of ubiquitin were detected from these samples, in which there was complete coverage of the ubiquitin amino acid sequence of 1-73”.

Based on this, we figure that the reason why mutations at Thr12 and Thr55 in Ub did not abolish the formation of hydroxyamine-sensitive bands is that there are alternative ubiquitination sites in the Ub Thr12/Thr55 mutant (such as Ser20 shown by Kelsall et al.), which we could not detect in our assays by using wild-type Ub.

4) The authors hypothesize that a key function of the HOIL-1 esterification activity is to form heterotypic chains. While this might be the case, the alternative hypothesis of HOIL-1 priming substrates with ester-linked ubiquitins that are then linearly extended by HOIP seems also equally valid, especially because multiple substrates have been reported to be modified with linear chains, but HOIP appears to be tailored to modify only an ubiquitin substrate. The authors should discuss this alternative hypothesis and why both systems might be important.In further support of substrates potentially being the most abundant ester-linked species, NEMO-enriched linear chains from TNF treated cells show a much more pronounced collapse compared to the ester-linked Ub-Ub linkages produced in the absence of substrate in vitro. It would greatly strengthen the paper if the authors could add a recombinant substrate (e.g. IRAK1/2 or MyD88) to the in vitro reaction.

The reviewers raise an important point concerning the architecture of the heterotypic ubiquitin chains and the LUBAC ligase responsible for substrate ubiquitination. Therefore, we carried out an in vitro ubiquitination reaction including the model LUBAC substrate NEMO and subjected the reactions to hydroxylamine treatment (Figure 6B). Our results show that the majority of the signal from the ubiquitinated substrate is unaffected by hydroxylamine, indicating that substrate-conjugated chains are branched but not attached to the substrate via oxyester bonds. We have discussed the in vivo and in vitro chain branching data as follows:

“Our analysis of chain branching on substrate-conjugated ubiquitin chains suggests that the oxyester branch points of the chain are located distally from the substrate. […] One explanation could be that there are other factors present in cells that assist HOIL-1L in the process of chain branching and which remain hitherto unidentified.”

5) In Figure 4B, why could the mixed LUBAC subunits generate a linear chain, but not an oxyester-linked branched Ub4? Do they form a high molecular weight complex that can be detected by gel filtration?

As the reviewers suggest, we believe that individually purified and mixed HOIP, HOIL-1L, and SHARPIN do not adequately assemble the LUBAC complex. The result being that, while HOIP auto-inhibition is released and linear chains can be assembled, the more catalytically complex action of HOIL-1L in chain branching is not possible. Perhaps this results from improper spatial arrangement of the RBR domains of HOIP and HOIL-1L resulting in interruption of the proposed Cys relay (Figure 8). To support this claim, we carried out mass photometry measurements from the individually purified LUBAC components mixed at an equimolar ratio of the three components, as was done for the comparative in vitro assay (Supplementary Figure 5 new). In line with our hypothesis we, did not observe the 220 kDa peak corresponding to assembled LUBAC nor the 440 kDa peak corresponding to dimerized LUBAC, indicating that mixed LUBAC components cannot adequately assemble the LUBAC complex.

6) In Figures 4E and 5A, it is interesting that Cezanne and vOTU could cleave ester-linked branched Ub4, although the molecular bases for these reactions remain unclear. Are the LUBAC-generated, hydroxylamine-sensitive His-Ub3 and Ub2 shown in Figure 5B cleavable by Cezanne and vOTU?

The reviewers ask an interesting question regarding the ability of the DUBs Cezanne and vOTU to cleave small ubiquitin polymers exclusively assembled with oxyester linkages. As suggested, we carried out a UbiCRest experiment of LUBAC-assembled His6-ubiquitin polymers. Our results show that the short oxyester-linked ubiquitin polymers are indeed sensitive to cleavage by both the tested DUBs (Supplementary Figure 6B new).

7) To confirm that the residual oligomeric Ub species after OTULIN treatment (Figure 4E) are exclusively ester-linked, a subsequent hydroxylamine treatment step should be performed.

We carried out OTULIN UbiCRest analysis of LUBAC-assembled poly-ubiquitin chains followed by hydroxylamine treatment, as the reviewers suggested (Supplementary Figure 6A new). Our results were consistent with a ubiquitin polymer containing predominantly M1 bonds connected to short branches via oxyester bonds. The majority of the poly-ubiquitin signal proved to be sensitive to cleavage by OTULIN, while a small residue of di- and tri-ubiquitin was resistant to OTULIN cleavage but degraded by subsequent treatment with hydroxylamine.

8) Finally, the suggestion that HOIP-HOIL Ub relay might be at play is exciting and implies that E3-mediated Ub relay might be a prevalent process. In principal it should be possible to test this by impairing E2 binding to the RING1 domain in HOIL in LUBAC. A steric mutation (e.g. X to Arg) would be a more elegant approach than a zinc-coordinating cysteine mutation. If relay is at play, then such mutated LUBAC should still be able to form ester linkages.

We agree with the reviewers that finding a way to impair E2 binding to HOIL-1L would be a great way to test the proposed Cys relay. Unfortunately, while we have tried to plan such an experiment, at present there are too many unknown variables to permit such a mutant from being made. The principal issue is with accurately predicting the E2 binding site of HOIL-1L. Some available structures of other RBR ligases such as HHARI and Parkin co-crystalized with an E2 enzyme show a binding loop on the RING1 domain (PDB: 5tte, 5udh, 6djw, 6n13). However, the only available structure of the HOIP RBR with an E2 (PDB: 5edv) shows a second E2-binding site at the RING2 and LDD domains. This raises the issue of whether a simple RING1 mutation in HOIL-1L would indeed prevent E2 binding by HOIL-1L or whether there is an additional E2 binding site at the RING2 domain. Furthermore, we carried out a multiple-sequence alignment of RBR family ligases and noted that a bulky hydrophobic residue is not conserved in HOIL-1L at the putative E2-binding loop. This leads to the possibility that if HOIL-1L binds the E2, it may bind it differently to the other RBR ligases. Therefore, without a high-resolution structure of the HOIL-1L RBR together with an E2 enzyme, a HOIL-1L E2-binding mutant cannot be reliably made. Failure to mutate all the correct residues could result in false positive or false negative results and would fail to address the model properly.